## [Transparent Peer Review file · Nature Communications]

Mosaic partial epidermal reprogramming remodels neighbors and niches to refine skin homeostasis and repair

Corresponding Author: Professor Sekyu Choi

Version 0:

Reviewer comments:

Reviewer #1

(Remarks to the Author)

The authors aim to explore partial reprogramming of skin epidermal cells using OSKM mice in which Yamanaka Transcription factors are expressed by epidermal cells. The authors report intriguing findings, including ectopic expression of keratin 17 and upregulation of upstream regulators such as phosphorylated AKT and EGFR. These may suggest an altered cell state in OSKM-expressing epidermal cells as well as secondary effects on other cells including neighboring keratinocytes, dermal cells, endothelial cells and T cells. Importantly, the authors demonstrate a significant enhancement of re-epithelialization in OSKM mice under both normal and diabetic conditions. The authors analyzed many distinct aspects of the phenotype in OSKM mice, making it a nice descriptive paper.

1. Their data do not convincingly demonstrate a loss of epidermal lineage identity. Traditionally, reprogramming via OSKM refers to the erasure of somatic cell identity and acquisition of pluripotency, enabling differentiation into multiple lineages. In the current study, it is unclear to what extent the cells undergo this transition. If the epidermal cells retain their original lineage characteristics, labeling the observed cellular changes as “reprogramming” may be misleading. Given that OSKM factors have diverse physiological functions depending on tissue context, the observed effects might result from another cellular responses rather than de-differentiation per se. Even at the expense of other experiments examining peripheral mechanisms in the paper, this fundamental issue needs to be much more robustly investigated with various methods.

2. Basically, the most important aspect of this study is to determine whether epidermal cells are really undifferentiated in this model under homeostasis and upon wounding. It is unclear whether K17 expression and K10 pattern distribution indicate de-differentiation of epithelial cells. Why epithelial cells with OSKM expression can still undergo terminal differentiation in the epidermis?

3. Can the authors describe how epithelial changes influence collagen composition? Is it known that epidermal differentiation promotes collagen I deposition? Given that the scarring depends upon dermal healing, more in depth mechanistic exploration would be helpful.

4. Hair follicle cells significantly contribute to skin physiology and wound healing. How are hair follicles altered in OSKM mice? Would the phenotype in OSKM mice partly due to the changes in hair follicle cells?

(Remarks on code availability)

Reviewer #2

(Remarks to the Author)

Kwak et al. demonstrate that transient expression of reprogramming factors in a subset of keratinocytes promotes skin repair through cell-autonomous and non-cell-autonomous mechanisms. The study is interesting, and the data are generally convincing.

The finding that partial reprogramming of cells promotes tissue repair is not completely novel and has been shown in other

tissues. However, this is the first study that shows the effect of partial keratinocyte reprogramming on epidermal homeostasis and wound repair. In particular, the finding that reprogramming promotes repair by cell-autonomous and non-autonomous mechanisms is interesting. However, the study has also several weaknesses and raises some questions (see specific comments below).

General points:

- 1.) The authors show interesting phenotypes after transient expression of OSKM in keratinocytes in mice. This is likely a consequence of partial reprogramming, but it cannot be excluded that the phenotype is simply a consequence of the expression of one of these transcription factors. Can the authors exclude this possibility?
- 2.) The translational relevance of the study is not very strong. It is unlikely that partial reprogramming will be used for the promotion of acute wound healing. The authors show the effect in diabetic mice, but this is a model for delayed acute healing, but not for chronic wounds. In a chronic wound, the risk of transformation is likely to be high, and the effect of reprogramming may be different than in an acute wound.

Specific points:

- 3.) Line 97: K14-CreER mice do not induce specific deletion in epidermal keratinocytes. The K14 promoter is also active in hair follicle keratinocytes and basal epithelial cells of other stratified epithelia. This sentence should be modified. In particular, it is unclear if and to what extent reprogrammed hair follicle keratinocytes contribute to the observed effects.
- 4.) The K14-CreER mice are known to be "leaky". The authors should include GFP and mCherry fluorescence pictures of mice that were not treated with Dox.
- 5.) In Fig. 1b the authors should add a high magnification hematoxylin/eosin staining of the skin. This is important to judge the overall histological phenotype of the mice.
- 6.) It is not clear what percentage of basal and suprabasal keratinocytes express OSKM. This information should be provided.
- 7.) Expression of Klf4, c-Myc and Oct-4 in the epidermis should also be confirmed (at least at the RNA level). This information should be available from the scRNA-seq data.
- 8.) The scRNA-seq data suggest alterations in apoptosis. Does reprogramming affect apoptosis of mCherry+ and mCherry- keratinocytes?
- 9.) The reversibility of the reprogramming effect is particularly important. Are the mCherry+ cells completely lost at day 10 after Dox withdrawal? If yes, do they undergo apoptosis or are they lost by terminal differentiation?
- 10.) Fig. 3g and h: PI3K inhibition is likely to affect epidermal cell proliferation in general. The authors should add control mice treated with the PI3K inhibitor for comparison. Same for the EGFR inhibitor in Fig. 4.
- 11.) The result in Fig. 3g requires quantification.
- 12.) Fig. 3j: It has not been shown that PI3K directly regulates keratin 6/16/17. This may be simply a consequence of the increased proliferation.
- 13.) Line 292: NRG (neuregulin) does not activate the EGFR – it activates ERBB3 and ERBB4.
- 14.) Fig. 4h: Please confirm the efficient EGFR inhibition by immunofluorescence staining for pEGFR.
- 15.) The model proposed in Fig. 4j is interesting, but it is only based on immunofluorescence stainings. It is important to test if PI3K activation in cultured keratinocytes promotes EGFR activation.
- 16.) Line 321: IL1 and VEGF do not directly regulate keratinocyte proliferation. This sentence should be modified.
- 17.) SFRP1 is a secreted protein, which should act on mCherry- and on mCherry+ cells. How do the authors explain a selective effect on mCherry- cells if SFRP1 is indeed responsible?
- 18.) Supplementary Fig. 5e: An important control is missing: LGK974 alone (same for the migration experiment). It seems likely that this compound already inhibits cell proliferation in control cells. The very weak effect of EGF on proliferation is surprising.
- 19.) Line 374: Epidermal T cells in mice are almost exclusively gd T cells. These cells are not recruited upon injury. They only populate the skin once during embryonic development (see publications from W. Havran and A. Hayday). The increase in these cells is likely a consequence of increased T cell proliferation, which should be tested. Alternatively, there may be an abnormal recruitment of ab T cells to the epidermis, which seems likely because there are more RORgt cells. Is there an

increase in epidermal and dermal ab or gd T cells or both?

20.) Lines 390-392: The effect of the CCR6 antagonist on the number of epidermal RORgt cells is not significant.

21.) Page 16, first paragraph: It is not surprising that no tumors were observed after 10 days of Dox treatment, because epidermal skin tumors develop slowly. It may well be that tumors would develop at a later time point (perhaps even after discontinuation of Dox treatment). This possibility should at least be discussed. In any case, it is clear from these results that long-term OSKM induction in keratinocytes is detrimental.

22.) The effect of partial reprogramming on wound closure is minor (Supplementary Figure S7A). This is not too surprising, because wound healing in healthy mice is highly optimized. The data on the length of the epithelial tongue and the transepidermal water loss are more convincing. It is essential to show histological stainings of cross-sections from wounds at different time points. Is the area of granulation tissue also affected?

23.) It is unclear if mCherry+ or mCherry- keratinocytes preferentially contribute to reepithelialization. This is mechanistically important and should be checked at different time points.

24.) The analysis of immune cells is restricted to T cells. However, macrophages are much more important players in wound healing. Is there a difference in the number of macrophages?

25.) Line 468: Fig. 6i,j and Supplementary Fig. 7a do not show that wound healing is delayed in the streptozotocin-treated mice. This should be shown in one figure. Again, histological cross-sections from the wounds at different stages should be shown. This is important to fully judge the phenotypic alterations.

26.) Line 494: It is correct that some reduction in the extent of angiogenesis can reduce scarring in mice. However, this is an indirect effect, which is not very strong. The extent of scarring strongly depends on myofibroblast differentiation. The authors should check if this is affected in the OSKM mice. It is mechanistically unclear how OSKM expression suppresses scarring. It may also be a consequence of the enhanced reepithelialization.

27.) Title of Fig. 1: The reprogramming does not drive epidermal dedifferentiation – it prevents differentiation (the K14 promoter targets non-differentiated basal cells).

(Remarks on code availability)

Version 1:

Reviewer comments:

Reviewer #2

(Remarks to the Author)

The authors performed a very extensive revision, and the revised manuscript is clearly improved.

Q1, Q2 and Q4 of reviewer 1 have been appropriately addressed, mainly by clarifying that epidermal cells undergo only a partial reprogramming, by further analysis of sc-RNA-seq data, and by further analysis of the hair follicles.

Q3 of reviewer 1 was partially addressed. The HIF1a inhibitor studies strongly suggest that HIF1a is required for wound angiogenesis (as expected from the literature) and also reduces dermal thickness. However, this was seen in control and in iOSKM mice. Therefore, the authors cannot conclude that HIF1a inhibition SPECIFICALLY reverses the iOSKM effect. The data on collagen III positive area suggest a specific effect of HIF1a inhibition on the iOSKM effect, but the variability of the data is high. I believe that this point requires further clarification in the manuscript. There are probably additional mechanisms that contribute to the effect of iOSKM on the dermis/granulation tissue.

It is difficult to explain how enhanced reepithelialization inhibits angiogenesis. Usually, it is the opposite, because keratinocytes are an important source of VEGF. This point should also be discussed.

Taken together, the (non-cell-autonomous) mechanisms underlying the alterations in dermal repair remain speculative, and therefore, this part of the study remains descriptive. This limitation should be mentioned in the discussion.

The questions of reviewer 2 have been appropriately addressed.

(Remarks on code availability)

REVIEWER COMMENTS

Reviewer #1 (Remarks to the Author):

The authors aim to explore partial reprogramming of skin epidermal cells using OSKM mice in which Yamanaka Transcription factors are expressed by epidermal cells. The authors report intriguing findings, including ectopic expression of keratin 17 and upregulation of upstream regulators such as phosphorylated AKT and EGFR. These may suggest an altered cell state in OSKM-expressing epidermal cells as well as secondary effects on other cells including neighboring keratinocytes, dermal cells, endothelial cells and T cells. Importantly, the authors demonstrate a significant enhancement of re-epithelialization in OSKM mice under both normal and diabetic conditions. The authors analyzed many distinct aspects of the phenotype in OSKM mice, making it a nice descriptive paper

We appreciate the reviewer's thoughtful assessment of our work and helpful suggestions, which have allowed us to further improve the quality and impact of our work.

Q1. Their data do not convincingly demonstrate a loss of epidermal lineage identity. Traditionally, reprogramming via OSKM refers to the erasure of somatic cell identity and acquisition of pluripotency, enabling differentiation into multiple lineages. In the current study, it is unclear to what extent the cells undergo this transition. If the epidermal cells retain their original lineage characteristics, labeling the observed cellular changes as "reprogramming" may be misleading. Given that OSKM factors have diverse physiological functions depending on tissue context, the observed effects might result from another cellular responses rather than de-differentiation per se. Even at the expense of other experiments examining peripheral mechanisms in the paper, this fundamental issue needs to be much more robustly investigated with various methods.

We appreciate the reviewer for this thoughtful suggestion, which we've addressed in our revised manuscript. The reviewer noted that the complete reprogramming of cells through OSKM expression typically involves the loss of the original cell identity and the acquisition of pluripotency. In contrast, our data indicate that "partial" epidermal reprogramming induces a transient and reversible plastic state, without progression to pluripotency. To avoid confusion, we have thoroughly revised the manuscript to clearly distinguish partial reprogramming from complete reprogramming.

(A) *In vivo* partial reprogramming refers to the induction of cellular plasticity without reaching pluripotency

Kurian et al. (*Nat Methods*, 2013) and Thier et al. (*Cell Stem Cell*, 2012) reported that transient *in vitro* expression of OSKM induces a dedifferentiated progenitor-like state rather than pluripotent stem cells. Specifically, Kurian et al. generated mesodermal progenitor cells from human fibroblasts, while Thier et al. generated neural stem cells, both demonstrating partial reprogramming without conferring full pluripotency^{1,2}.

Following these findings, Ocampo et al. (*Cell*, 2016) and Browder et al. (*Nat Aging*, 2022) demonstrated that short-term OSKM induction *in vivo* ameliorates age-associated hallmarks and prevents the loss of cell identity, tumorigenesis, and functional decline, thereby extending lifespan in a mouse model of accelerated aging^{3,4}. Ocampo et al. emphasized that "reprogramming proceeds in a stepwise manner allowing for the induction of partial reprogramming without the complete loss of cellular identity by short exposure to the Yamanaka factors". More recently, Wang et al. (*Nat Commun*, 2021) reported that *in vivo* partial reprogramming of myofibers promotes muscle regeneration, demonstrating that *in vivo*

partial reprogramming, while preserving the original identity, can be physiologically beneficial⁵. Furthermore, Chen et al. (*Science*, 2021) reported partial reprogramming of cardiomyocytes (CMs), noting that “transient expression of OSKM in CMs stimulates proliferation without loss of CM identity”, and demonstrated this approach promotes cardiac repair. They referred to this process as “partial reprogramming” or “reversible reprogramming”⁶. Similarly, Kim et al. (*Sci Adv*, 2023) showed short-term OSKM expression enables intestinal regeneration by transient dedifferentiation without loss of their original identity⁷. Yucel et al. (*Nat Commun*, 2024) summarized that “partial reprogramming aims to maintain cellular identity while achieving cellular rejuvenation, whereas full iPSC reprogramming aims to facilitate cellular dedifferentiation”⁸.

Collectively, these studies demonstrate that the concept of “partial reprogramming” is distinct from the traditional “complete reprogramming”, as it does not progress to pluripotency. Our work follows this established definition of partial reprogramming.

(B) Our data demonstrates partial epidermal reprogramming, which induces epidermal lineage-restricted cellular plasticity with regenerative features

We confirmed that OSKM expression in epidermal cells for 3 or 9 days did not induce Nanog expression (Supplementary Fig. 2a and 12b), indicating that complete reprogramming did not occur. By contrast, long-term OSKM induction for 2 months in epidermal keratinocytes resulted in Nanog expression and teratoma formation *in vivo* (Rebuttal Fig. 1), showing that the epidermal cells can acquire pluripotency under long-term induction. These findings suggest that partially reprogrammed cells induced by short-term (3 day) OSKM expression include intermediate reprogrammed populations that remain on the complete reprogramming trajectory.

Nanog was not induced under short-term epidermal OSKM expression for 3 or 9 days (Supplementary Fig. 2a and 12b)

Nanog was induced under long-term epidermal OSKM expression for 2 months (Rebuttal Fig. 1)

Although pluripotency was not induced, our multiple lines of data support the acquisition of cellular plasticity and a decrease in differentiated keratinocyte characteristics following transient OSKM expression. Cytoscape2 analysis revealed increased cell potency after 3-day OSKM expression (Supplementary Fig. 7e). Expression of Krt17, Krt16, and Krt6a, representative hallmarks for epidermal plasticity⁹⁻¹¹, was upregulated in the partially reprogrammed epidermis (Fig. 2g-i, n; Supplementary Fig. 6d). In addition, epidermal

differentiation markers including Krt10 were reduced in suprabasal keratinocytes (Fig. 1f-h; Supplementary Fig. 4c,d). Pseudotime and RNA velocity analyses of our scRNA-seq data further showed that suprabasal keratinocytes in Epi-iOSKM epidermis displayed reduced differentiation signatures and exhibited cell fate trajectories opposite to normal epidermal differentiation (Fig. 1i,l-n; Supplementary Fig. 4f-h,k-n). In addition, multiple datasets indicated increased proliferation in the epidermis (Fig. 1j,k; Supplementary Fig. 4i,j). To more clearly illustrate cellular transition into more plastic state with reduced differentiation characteristics, we incorporated additional scRNA-seq analyses in the revised manuscript (new Fig. 1m,n; new Supplementary Fig. 4d,g,h,l-n; see also responses to reviewer 1, question #2).

Partial epidermal reprogramming confers cellular plasticity (Supplementary Fig. 7e; Fig. 2n)

Partial epidermal reprogramming attenuates epidermal differentiation characteristics (Fig. 1i, l)

Partial epidermal reprogramming induces cell proliferation (Fig. 1k)

Additionally, we found that after 3 days of short-term partial reprogramming, epidermal cells preserved their original identity (Supplementary Fig. 7i-k) while acquiring plasticity with regenerative features (Fig 2; Supplementary Fig. 7e). This state was reversible (Supplementary Fig. 7f-h), indicating that epidermal keratinocytes subjected to transient OSKM expression *in vivo* were partially reprogrammed within their epidermal cell lineages.

Taken together, these results provide molecular and functional evidence that transient OSKM expression partially dedifferentiates epidermal keratinocytes with preserving their lineage identity (see also responses to reviewer 1, question #2), consistent with the widely accepted definition of partial reprogramming as loss of differentiation markers and increased proliferation without acquisition of pluripotency.

(C) Injury-responsive-like conversion and enhanced healing outcomes in Epi-iOSKM skin result from partial reprogramming-induced epidermal plasticity

Importantly, the restricted cellular plasticity within epidermal cell lineage upon ectopic OSKM expression recapitulated the natural partial dedifferentiation process during wound repair (Fig.

2). Integrative analysis of our scRNA-seq data with reference datasets showed that partially reprogrammed cells and wounded cells share common cellular characteristics (Fig. 2b-e). For example, not only Krt17 and Krt6 were induced, but also stress responses such as HIF-1 α activation were induced in partially reprogrammed cells and their neighbors, similar to the wound repair process (Fig. 2f-m). Plastic cells induced by wounding propagate injury responses to their surroundings for rapid restoration of the tissue^{12,13}. Similarly, plastic IFE cells induced by short OSKM induction extended partial reprogramming effects to their neighbors and niches through hijacking EGFR signaling and CCL20-CCR6 axis that crucially function during wound repair (Fig. 4,5). These collective changes improved skin wound repair in Epi-iOSKM mice (Fig. 6, new Fig. 8). Notably, our new data revealed that epidermis-wide HIF-1 α activation upon mosaic partial reprogramming, one of the recapitulative phenotypes of healing characteristics¹⁴, was a key modulator of enhanced repair in Epi-iOSKM mice (new Fig. 7i-n, see also responses to reviewer 1, question #3). Together, these findings suggest that mosaic partial epidermal reprogramming, by conferring plasticity within epidermal cell lineages, induces regenerative features not only in themselves but also in their neighbors and niches to enhance skin repair.

Partial reprogramming-induced epidermal plasticity confers regenerative characteristics to themselves and their neighbors/niches (Fig. 9a, images are reproduced with permission from Bioinsight under the CC BY license)

Ultimately, the major phenotypes observed in Epi-iOSKM epidermis—increased proliferation, loss of differentiated characteristics, acquisition of regenerative characteristics, and the extended effects of OSKM-expressing cells on their neighbors and niches—represent direct and indirect outcomes of partial reprogramming, which confers cellular plasticity within epidermal lineages.

As a result, the Epi-iOSKM epidermis collectively exhibited injury-responsive-like characteristics (Fig. 2; Supplementary Fig. 7), resembling a state in which healing has already been initiated before wound occurrence. This led us to hypothesize that partial epidermal reprogramming would facilitate wound repair. Consistent with this, we observed accelerated re-epithelialization and reduced scarring. In summary, partial reprogramming confers plasticity to epidermal cells within the lineage, enabling them to mimic repair processes, which proved broadly beneficial for cutaneous healing.

(D) Corrections and clarifications in the revised manuscript

As noted at the outset, and supported by both our own data and reference studies, the

observations in this study reflect the effects of “partial reprogramming”, not “reprogramming” in the traditional concept of full reprogramming. However, in the original manuscript, the two terms were occasionally used interchangeably, which may have led to confusion.

To address this, we have systematically revised the terminology throughout the manuscript. For example:

- epidermal reprogramming → partial epidermal reprogramming
- reprogrammed cells/epidermis → partially reprogrammed cells/epidermis

In line with this correction, we also revised the paper title to: Mosaic partial epidermal reprogramming remodels neighbors and niches to refine skin homeostasis and repair.

The titles of Fig. 3 and Fig. 4 were also revised as follows:

- Fig. 3: Partially reprogrammed epidermal cells recapitulate repair states via PI3K activation.
- Fig. 4: Non-cell-autonomous effects in the partially reprogrammed epidermis via EGFR activation.

In addition, we have added an illustration in the Introduction to clearly explain the concept of partial reprogramming, as shown below:

“Cellular reprogramming by Oct-4, Sox2, Klf4, and c-Myc (OSKM), known as the Yamanaka factors, has emerged as a valuable tool for the manipulation of cell fate. Originally, OSKM were used only in vitro because their forced expression in vivo leads to teratoma formation due to the production of induced pluripotent stem cells. However, it was found that transient and reversible OSKM expression could induce “partial reprogramming”, which confers cellular plasticity while retaining the original cell identity and functionality. This process did not induce pluripotency, indicating that in vivo partial reprogramming could prevent teratoma formation. Partial reprogramming in mice rejuvenated progeria or physiological aging. Furthermore, recent studies have revealed that in vivo partial reprogramming promotes tissue repair in various organs, such as the muscle, liver, and intestine. Another study observed a reduction in fibrosis and scarring after injury.”

Taken together, the more detailed description of reference studies, our original and newly provided data, along with clarification of ambiguous wording, and the addition of explanations of the key concept in the revised manuscript collectively demonstrate that our study is the first to report both cell-autonomous and non-autonomous effects of *in vivo* “partial reprogramming” of epidermal cells on skin homeostasis and repair.

Q2. Basically, the most important aspect of this study is to determine whether epidermal cells are really undifferentiated in this model under homeostasis and upon wounding. It is unclear whether K17 expression and K10 pattern distribution indicate de-differentiation of epithelial cells. Why epithelial cells with OSKM expression can still undergo terminal differentiation in the epidermis?

Thank you for this important comment. We do not claim that epidermal cells become “undifferentiated” in the sense of losing epidermal lineage identity or acquiring pluripotency. Throughout the revised manuscript, we use “partial reprogramming” in the established meaning: a transient, reversible increase in cellular plasticity within the epidermal cell lineage, without progression to pluripotency. Please see also responses to question #1.

Krt17 induction and Krt10 pattern distribution represent hallmarks of this partial dedifferentiation of epidermal cells with preserving their own lineage, contrasting with full dedifferentiation into iPSCs. Importantly, these marker alterations are only part of the broader phenotypes that support this process. In addition to them, extensive original and additional

data demonstrate epidermal lineage-restricted dedifferentiation:

1. No pluripotency program under short-term OSKM expression: Short-term OSKM induction (3 or 9 days) did not induce Nanog expression (Supplementary Fig. 2a, 7i, 12b), and cells retained epidermal lineage markers (*Krt5* and *Trp63*, Supplementary Fig. 7j,k) while resuming the normal differentiation cycle after Dox withdrawal (Supplementary Fig. 7f-h).

2. Differentiation characteristics are attenuated, not abolished, and reversible: *Krt10*, a marker for epidermal differentiation, in suprabasal cells was reduced (not absent) under short-term OSKM expression and recovered after Dox withdrawal (Fig. 1f-h; Supplementary Fig. 7h), consistent with a reversible plastic state rather than lineage erasure. Cleaved caspase-3 staining was negative during and after induction (new Supplementary Fig. 2c,d), indicating recovery occurred without apoptotic loss. In addition to the marker gene expression, pseudotime analysis with scRNA-seq showed that comprehensive epidermal differentiation extent was declined (Fig. 1i; new Supplementary Fig. 4g,h). We also observe the similar pattern of decrease in another suprabasal marker *Dsg1a* in the new manuscript (new Supplementary Fig. 4d).

Partial reprogramming reduces epidermal differentiation markers (Fig. 1g; new Supplementary Fig. 4d) and correlated pseudotime values (Fig. 1i; new Supplementary Fig. 4g,h)

3. Cell property reversion toward an earlier state (Supra(*Krt10*⁺) → Basal(*Krt14*⁺)): scRNA-seq RNA velocity analysis in the previous manuscript exhibited that cell fate trajectories from Supra to Basal cells, suggesting a reversal of epidermal differentiation upon partial epidermal reprogramming (Fig. 1l). In the new manuscript, we additionally examined the marker gene expression and identified that fate transition trends of Epi-iOSKM IFE cells were correlated with decreases in differentiation markers and increases in epidermal stem cell markers. Specifically, the trends were from *Krt10*/*Dsg1a*-high to *Krt10*/*Dsg1a*-low and *Krt14*/*Itga6*-high states, exhibiting the opposite trend from the normal epidermal differentiation. We have added this result as new Fig. 1m and new Supplementary Fig. 4l-n.

Cell fate transition from Supra to Basal cells under partial reprogramming (Fig. 1l)

Cell fate transition from differentiated to stem-like IFE cells under partial reprogramming (new Fig. 1m; new Supplementary Fig. 4l-n)

4. Repair-like, lineage-restricted plasticity: Integrative scRNA-seq analysis with public reference wound datasets revealed that Epi-iOSKM IFE cells acquire the epidermal plasticity that deviates toward a wound-responsive state, not toward a pluripotency program (Fig. 2b-e). Upregulation of Krt17/Krt6 and stress-response pathways such as HIF-1 α and STAT3 is a hallmark of activated/wound-edge keratinocytes, not of full dedifferentiation (Fig. 2f-m).

Together, these findings explain why OSKM-expressing IFE cells can still undergo terminal differentiation *in vivo*: partial reprogramming confers lineage-restricted, reversible plasticity that primes cells for repair without extinguishing the epidermal differentiation network. After OSKM withdrawal, epidermal architecture and marker expression normalize, and no pluripotency markers are detected.

We have clarified this definition and supporting evidence in the Introduction and Results, and we now consistently use the term “partial reprogramming” throughout to avoid confusion.

Q3. Can the authors describe how epithelial changes influence collagen composition? Is it known that epidermal differentiation promotes collagen I deposition? Given that the scarring depends upon dermal healing, more in depth mechanistic exploration would be helpful.

We appreciate this comment, which prompted us to further investigate a mechanistic link between partial epidermal reprogramming and enhanced dermal healing. Through additional experiments, we identified epidermal HIF-1 α activation as a key factor modulating enhanced dermal healing in Epi-iOSKM skin (new Fig. 7i-n; new Supplementary Fig. 15).

Re-epithelialization can influence collagen deposition in multiple ways, but due to the complexity of the wound healing process, its effects cannot be generalized and may vary depending on context. For example, accelerated re-epithelialization can reduce scar formation by shortening the overall repair duration and preventing excessive healing¹⁵⁻¹⁷, but in other cases, rapid epithelial closure can instead promote increased scarring by driving overactive healing processes¹⁸⁻²⁰.

In the Epi-iOSKM skin, accelerated re-epithelialization was accompanied by reduced scarring, indicating enhanced repair in both epidermis and dermis. Because mosaic partial reprogramming of the epidermis was stopped at 1 day post-wounding (dpw)—a very early stage of the healing process—these long-lasting beneficial phenotypes likely represent

extended effects from a temporal perspective as well as from a spatial perspective. In the original manuscript, we reported that HIF-1 α activity is more elevated in Epi-iOSKM epidermis at 3 and 7 dpw, even after the partial reprogramming had already ceased, unlike in unwounded skin (Fig. 7b-e). This suggests that HIF-1 α may mediate the prolonged effects of reversible reprogramming.

Notably, HIF-1 α was activated in the Epi-iOSKM epidermis in an atypical pattern: more increased activation was detected in the old epidermis that had undergone partial reprogramming, rather than in the newly regenerated epithelium (Fig. 7b-d). Interestingly, angiogenesis patterns during wound repair were significantly altered in Epi-iOSKM skin according to the altered pattern of HIF-1 α activation in the epidermis upon partial reprogramming (Fig. 7f-h), suggesting that HIF-1 α may also spatially extend the effects of partial epidermal reprogramming to the dermal repair process.

Partial epidermal reprogramming increases HIF-1 α activity and alters neovascularization patterns during wound repair (Fig. 7a-h)

To test this, we examined the role of epidermal HIF-1 α activation in the wound healing process of Epi-iOSKM mice. We found that HIF-1 α is essential not only for promoting re-epithelialization but also for reducing scar formation (new Fig. 7i-n; new Supplementary Fig. 15a-e). Together, these findings demonstrate that partial epidermal reprogramming promotes dermal collagen remodeling and scar attenuation through activation of epidermal HIF-1 α .

The related data and explanation in the revised manuscript are as follows:

“Previous studies have reported that forced activation of HIF-1 α accelerates re-epithelialization. In our model, HIF-1 α activation induced by partial reprogramming was sustained after wounding (Fig. 7c), suggesting it may continue to promote re-epithelialization even after transient reprogramming has ceased. Moreover, the altered patterns of HIF-1 α activation correlated with reorganized angiogenesis (Fig. 7b-h), potentially influencing dermal repair. Notably, appropriate regulation of neovascularization during wound repair is known to reduce scar formation.

When a wound occurs in Epi-iOSKM mice, HIF-1 α is activated twice in the epidermis—first by partial reprogramming and again by the wound itself. To disrupt the synergistic effect of these

two stimuli, we transiently inhibited HIF-1 α activity by treating the HIF-1 α inhibitor PX-478 once at the time of wounding (Fig. 7i). HIF-1 α activity decreased at 1 dpw but recovered by 10 dpw as expected (Supplementary Fig. 15a,b). Interestingly, transient HIF-1 α inhibition during the early repair phase reversed the accelerated re-epithelialization in Epi-iOSKM mice, bringing it to levels comparable to controls (Fig. 7j,k). Moreover, temporal inhibition of HIF-1 α markedly reduced angiogenesis. Without inhibition, control mice displayed widespread angiogenesis across the wound bed, whereas Epi-iOSKM mice showed angiogenesis concentrated near the original wound site (Fig. 7f-h). Following PX-478 treatment, however, both groups exhibited a pronounced reduction in angiogenesis, with low CD31 expression at newly formed dermis near both the old wound edges and the regenerated epidermis (Fig. 7j,l; Supplementary Fig. 15c,d). These findings indicate that HIF-1 α is a key regulator of neovascularization patterns during wound repair in both control and Epi-iOSKM mice. We next examined whether HIF-1 α , which is essential for the alterations in re-epithelialization and angiogenesis in Epi-iOSKM mice, also contributes to scar reduction. At 20 dpw, dermal thickness in the scarred area of PX-478 treated Epi-iOSKM mice was significantly thicker than that of untreated Epi-iOSKM mice and was comparable to that of PX-478 treated control mice (Fig. 7m). Consistently, the alterations in collagen I deposition observed in Epi-iOSKM skin were restored by PX-478 treatment (Supplementary Fig. 15e). In addition, the reduced proportion of collagen III+ area observed in Epi-iOSKM mice was reversed by HIF-1 α inhibition during partial epidermal reprogramming (Fig. 7n), indicating that elevated HIF-1 α levels are essential for the reduction in scarring. Collectively, increased HIF-1 α activation upon partial epidermal reprogramming—particularly the atypical pattern observed in the old epidermis near the initial wound site rather than in the newly generated epithelium (Fig. 7b-d)—substantially contributes to the widespread changes that promote wound repair.”

HIF-1α activation is essential for enhancing wound healing and attenuating scar formation in Epi-iOSKM skin (new Fig. 7i-n; new Supplementary Fig. 15a-e)

These results illustrate how epidermal changes can influence dermal healing. HIF-1α, a responder to stressful environments, is rapidly activated after wounding and coordinates multiple repair processes^{12,21}. When the control group was transiently treated with by a HIF-1α inhibitor, re-epithelialization was not altered, but dermal healing process was affected (new Fig. 7j-l), suggesting that the primary role of HIF-1α during the early phase of wound healing is to coordinate dermal niche activation. Thus, the alterations in HIF-1α activity may broadly affect healing processes. Indeed, spatial and temporal regulation of HIF-1α is known to exert complex, context-dependent effects on healing, which can be either beneficial or detrimental^{14,21,22}.

The Epi-iOSKM epidermis exhibited the elevation of HIF-1α activity as an unusual pattern: the signal was increased in the old epithelium that had undergone partial reprogramming, rather than in the newly regenerated epithelium (Fig. 7b-d).

HIF-1 α activation in wounded Epi-iOSKM epidermis (Fig. 7b-d)

This alteration substantially remodeled wound repair mechanisms, including re-epithelialization, angiogenesis, and scarring (Fig. 9b). Given the complexity of the changes, it is challenging to establish direct causality among the different repair processes. Nevertheless, our results suggest the potential that accelerated re-epithelialization, accompanied by reduced neovascularization near the rapidly regenerating epithelium, contributes to decreased scar formation. Importantly, HIF-1 α emerges as a key regulator coordinating these processes. Taken together, partial epidermal reprogramming, which induces healing characteristics, activated HIF-1 α even prior to wounding (Fig. 2l), and wounding itself also activated HIF-1 α . Consequently, the Epi-iOSKM epidermis during healing exhibited an atypical pattern of HIF-1 α activation, which appears to promote coordinated repair of both the epidermis and dermis.

Q4. Hair follicle cells significantly contribute to skin physiology and wound healing. How are hair follicles altered in OSKM mice? Would the phenotype in OSKM mice partly due to the changes in hair follicle cells?

Thank you for the valuable comments. We agree that hair follicular cells can contribute to skin physiology and the wound healing process^{23,24}. However, in our Epi-iOSKM mice, hair follicle cells exhibited markedly low OSKM induction efficiency and showed minimal phenotypic changes, suggesting that they contributed little to the observed outcomes.

Although the Krt14 promoter does not induce an entirely specific induction, both our observations and previous reports indicate that the relative efficiency of gene induction is substantially higher in the interfollicular epidermis (IFE) than in hair follicles when using Krt14-driven CreER systems (Jax #005107)²⁵. In our model with the administered tamoxifen (TAM) and DOX dosages, we consistently observed that TAM-induced GFP expression and the subsequent mCherry expression were much more robust in IFE cells than in the hair follicle cells (new Supplementary Fig. 1b,c). Consistently, transient OSKM expressions induced under the Krt14 promoter did not alter the proliferative capacity of hair follicle cells (Rebuttal Fig. 2), whereas cell proliferation was markedly increased in IFE cells (Fig. 1j,k).

Fluorescence induction in hair follicles of Epi-iOSKM mice was detected only at very low levels compared to that in the interfollicular epidermis (new Supplementary Fig. 1b,c)

No detectable changes were observed in hair follicles of Epi-iOSKM mice (Rebuttal Fig. 2)

Despite the low efficiency of OSKM-mCherry expression in hair follicle cells, we examined whether hair follicle cells influence the phenotypes observed in the Epi-iOSKM skin. As expected, the expression of EGF ligands (*Tgfa*, *Epgn*, and *Areg*), which can induce non-cell-autonomous effects in mCh^{neg} IFE keratinocytes, as well as immune cytokines (*Tnf*, *Ccl8*, *Cxcl10*, *Ccl20*, *Il1b*, and *Cxcl16*), which can induce T cell recruitment and activation, were not significantly altered in Epi-iOSKM hair follicle cells (Rebuttal Fig. 3). These results suggest that hair follicles had little influence on the phenotypes we reported in our study.

Hair follicles of Epi-iOSKM mice show no detectable alterations (Rebuttal Fig. 3)

Consistently, K14-CreER mice driven by the human K14 promoter have been widely used in studies specifically examining the characteristics of the IFE²⁶⁻²⁸. Moreover, Zhang et al. (*Cell Cycle*, 2010) showed that among several Krt14-CreER strains, the line used in our study (Jax #005107) exhibits relatively low sensitivity to tamoxifen and highly inefficient recombination within hair follicles, whereas K14-CreER2 (Jax #038390) is a more suitable model for studying both hair follicles and the epidermis²⁵. In addition, because we induced OSKM expression using the K14-CreER line (Jax #005107) in combination with the Dox-inducible system, the observed activity may have been concentrated in the more efficient IFE. Taken together, while some recombination in hair follicle cells is expected, both our data and prior reports demonstrate that the human K14 promoter functions most strongly and consistently in the epidermis. Thus, in the context of our study, the induced genetic changes predominantly reflect biology in the epidermis rather than in hair follicles (please see also responses to Reviewer 2, question #3).

Inefficient recombination was observed within hair follicles when using the same K14-CreER driver in our system (Zhang et al., *Cell Cycle*, 2010)²⁵

Reviewer #2 (Remarks to the Author):

Kwak et al. demonstrate that transient expression of reprogramming factors in a subset of keratinocytes promotes skin repair through cell-autonomous and non-cell-autonomous mechanisms. The study is interesting, and the data are generally convincing.

The finding that partial reprogramming of cells promotes tissue repair is not completely novel and has been shown in other tissues. However, this is the first study that shows the effect of partial keratinocyte reprogramming on epidermal homeostasis and wound repair. In particular, the finding that reprogramming promotes repair by cell-autonomous and non-autonomous mechanisms is interesting. However, the study has also several weaknesses and raises some questions (see specific comments below).

We appreciate the reviewer's positive assessment and helpful suggestions, which helped us to further enhance the quality and impact of our study.

General points:

Q1. The authors show interesting phenotypes after transient expression of OSKM in keratinocytes in mice. This is likely a consequence of partial reprogramming, but it cannot be excluded that the phenotype is simply a consequence of the expression of one of these transcription factors. Can the authors exclude this possibility?

Thank you for the thoughtful comments and suggestions. Investigating the contribution of individual transcription factors (TFs) to cellular reprogramming or partial reprogramming is indeed an important question, particularly for identifying the minimal TF combination that can confer plasticity. However, as this work represents the first investigation of mosaic partial reprogramming in the skin epidermis, our primary objective was to comprehensively characterize its phenotypes across multiple skin cell types during homeostasis and repair, including the complex interplays between partially reprogrammed cells and their neighbors/niches, rather than to dissect the role of each factor. While defining the contribution of each factor is meaningful for identifying molecular mechanisms of partial reprogramming and optimal utilization conditions, we believe our work, which has deeply and broadly elucidated the fact that their combination induces intriguing and valuable phenotypes, is already sufficiently comprehensive. Moreover, recent studies support that the effects of OSKM are largely mediated through their combined action.

(A) Importance of the collective effects of OSKM in reprogramming

Recent studies on partial and full reprogramming consistently indicate that no single Yamanaka factor alone can account for the broad phenotypes of reprogramming. To be specific, Sheng et al. (*Nat Commun*, 2018) reported that Oct4 is not essential for epigenetic rejuvenation under partial reprogramming²⁹; likewise, Onder et al. (*Nature*, 2012) showed that Klf4 is not necessary for achieving reprogramming³⁰. Furthermore, Sox2 plays a crucial role in full reprogramming as part of multimeric complexes with Oct4 and Klf4, rather than functioning alone³¹. Lu et al. (*Nature*, 2020) found that c-Myc is dispensable for *in vivo* partial reprogramming in some contexts³². Yuçel et al. (*Nat Commun*, 2024) emphasized that epigenetic reprogramming is not attained through overexpression of a single factor, but rather through synergistic interactions among multiple factors and their downstream networks⁸. Taken together, these reports support that the phenotypes we observed are unlikely to be explained by a single factor, but rather by the combined action of OSKM.

(B) c-Myc is not a main contributor to the induction of healing features in the Epi-iOSKM epidermis

Although definitive verification of each factor's role in the epidermis, an approach that would require extensive resources and future studies, we performed exploratory experiments within the feasible scope of our laboratory. We introduced viral constructs carrying Oct4, Sox2, Klf4, or c-Myc sequences individually into the skin dermis of mice, with the intention of assessing whether any single factor could induce regenerative features, the most crucial phenotypes observed in the Epi-iOSKM epidermis. However, as previously reported, the delivery efficiency to the epidermis was extremely low^{33,34}, limiting meaningful analyses.

As an alternative approach, we conducted the inhibition of a specific factor during partial epidermal reprogramming. In particular, we focused on c-Myc, which has been reported in certain other tissues as a non-essential factor for *in vivo* partial reprogramming while synergistic effects among OSK are still necessary³². To test whether c-Myc is dispensable in our model, we treated control and Epi-iOSKM skin with the c-Myc inhibitor 10058-F4³⁵. We found that Krt17/6a induction and HIF-1 α activation were maintained in the Epi-iOSKM+10058-F4 epidermis (Rebuttal Fig. 4). These results indicate that c-Myc is not essential for the healing features induced by partial epidermal reprogramming. Although we cannot entirely exclude the contribution of c-Myc in mosaic partial epidermal reprogramming effects, our results demonstrated that the healing features can also be induced by the combination of Oct4, Sox2, and Klf4.

Induction of healing features in Epi-iOSKM epidermis is maintained under c-Myc inhibition (Rebuttal Fig. 4)

Although delineating the precise contribution of each OSKM factor will be an important avenue for future studies, our experimental results—together with recent reports—provide evidence that the phenotypes in Epi-iOSKM mice arise from the cooperative and synergistic actions of reprogramming factors in the epidermis.

Q2. The translational relevance of the study is not very strong. It is unlikely that partial reprogramming will be used for the promotion of acute wound healing. The authors show the effect in diabetic mice, but this is a model for delayed acute healing, but not nor chronic wounds. In a chronic wound, the risk of transformation is likely to be high, and the effect of reprogramming may be different than in an acute wound.

Thank you for the comments. They allowed us to refine the potential practical applications of our work and, with additional experimental data, to better address the translational relevance of partial epidermal reprogramming to promote wound healing.

(A) Clinical implications of our data demonstrating enhanced healing in acute wounds

The reviewer noted that our data are restricted to normal and delayed acute wounds, rather than chronic wounds. However, we believe our findings, which clearly demonstrate improved acute wound healing, are also of practical clinical relevance. Most human wounds are acute, but delayed healing can lead to secondary complications such as infection, which in turn may progress to chronic wounds^{16,17}. Therefore, it is essential to treat acute wounds at risk of delayed healing promptly to prevent progression to chronicity. This is particularly important in surgical wounds, where delays in repair can cause more severe secondary damage. Accelerating acute healing is valuable in a broader clinical context, as it improves overall

recovery and reduces the risk of long-term complications.

In our study, we employed a silicone-splinted excisional model to minimize contraction and promote re-epithelialization-dominant closure³⁶, thereby enhancing the translational relevance to human wound healing. Importantly, beyond the direct application of partial reprogramming, our data also provide mechanistic insights for developing new therapeutics for injury repair. We have shown that transient activation of epidermal cells at early stages of healing can be beneficial for tissue repair. This finding offers new avenues for developing pharmaceuticals aimed at enhancing wound healing, as discussed in the revised manuscript: *“From a translational perspective, our findings possess the potential that not only OSKM expression but also other ways that results in epidermal plasticity in a mosaic and transient manner could enhance wound healing in a similar manner”*. Particularly, we have demonstrated that partial reprogramming of the epidermis also contributes to enhanced dermal healing. This represents a novel and meaningful phenotype, as we have discussed in the manuscript: *“This suggests that epithelial stem cell progenies act as a key player regulating the overall skin repair process, unlike most previous studies that focused on how these cells were regulated by their niches”*.

In particular, our new data demonstrate that increased epithelial HIF-1 α activation is essential for enhancing not only epithelial but also dermal healing (new Fig. 7i-n; new Supplementary Fig. 15). This finding provides specific insights into how transient changes in the epidermis during the early stages of wound repair can facilitate dermal restoration and reduce scarring (please see also responses to Reviewer 1, question #3, and Reviewer 2, question #26). Reduced scarring is further associated with complete regeneration, which is characterized by the absence of scar formation and the induction of hair follicle neogenesis^{24,37,38}; so, our findings hold significant potential to serve as a foundational basis for extensive practical applications.

Collectively, we believe that the direct application of partial epidermal reprogramming may be feasible in certain contexts. Beyond this, our findings—highlighting the important roles of epithelium in orchestrating wound healing—should be valuable for researchers studying tissue repair and regeneration. Exploring other factors that can recapitulate the effects of partial reprogramming by targeting epidermal cells would be a promising avenue for future studies with translational potential.

(B) Clinical implications of our data demonstrating enhanced healing in diabetic wounds with defective repair

Although our findings already hold clinical value, as the reviewer noted, including data on chronic wounds would further strengthen the translational relevance. However, establishing a truly chronic, non-healing wound model in mice remains highly challenging. Numerous mouse studies attempting to improve chronic wound healing have established wound conditions using various factors known to promote wound chronicity such as diabetes, oxidative stress, or infection. However, most attempts produced delayed wound healing rather than non-healing wounds³⁹⁻⁴³. There is no rodent model that fully replicates human chronic wounds that do not heal^{39,44-46}. Nevertheless, fortunately, frequently utilized defect healing models in laboratories, such as the hyperglycemia mice, still provide highly translational relevance by enabling the study of potential therapeutics targeting key drivers of severe impairments in wound healing^{39-43,47}. Consistent with this rationale, we employed the streptozotocin (STZ)-induced hyperglycemia mouse model (new Fig. 8; new Supplementary Fig. 16), the most widely used impaired repair model, to investigate the effects of partial epidermal reprogramming on defective wound healing.

In addition to our original data, and in response to question #25 regarding validation of this model, we further characterized the diabetic wound healing process (new Fig. 8; new Supplementary Fig. 16). We then identified several previously known wound healing disorder

phenotypes induced by hyperglycemia⁴⁸⁻⁵⁰ in our model. Healing was not only delayed (revised Fig. 8b,d,e), but the proliferative ability of epithelial cells was also markedly reduced in STZ-treated mice (new Fig. 8c). Furthermore, the angiogenesis and dermal healing were significantly impaired (new Fig. 8d,f; new Supplementary Fig. 16d). These new results demonstrate that our diabetic model exhibits not merely delayed healing, but the defective healing with pathophysiological alterations.

In this defect model, partial epidermal reprogramming exhibited broad beneficial effects. Wound closure rates increased (revised Fig. 8b), and re-epithelialization was accelerated (revised Fig. 8d,e). Moreover, the proportion of proliferating epidermal keratinocytes were increased at 3 days post-wounding (dpw) (new Fig. 8c). In wildtype mice with no diabetes, epidermal proliferation was affected by partial reprogramming only under homeostatic conditions, but not during repair (new Supplementary Fig. 14b), suggesting partial epidermal reprogramming effects are distinct in diabetic mice compared with those in wildtype mice. Further, distinctive effects of partial epidermal reprogramming on diabetic models were observed also in dermal healing. Unlike in wildtype mice, where partial epidermal reprogramming only altered the pattern of angiogenesis, in the diabetic model where angiogenesis was sharply reduced, partial epidermal reprogramming restored the diminished angiogenesis (new Fig. 8d,f). Reduced granulation tissue area by STZ treatment was also rescued by partial epidermal reprogramming (new Supplementary Fig. 16d). These results suggest that while the effects of partial epidermal reprogramming act differently in a diabetes model exhibiting dysfunctional repair processes, it ultimately improves wound healing in a positive direction even in disease models.

Improvement of diabetic wound healing by partial epidermal reprogramming (new Fig. 8)

Collectively, together with our observations that STZ treatment results in impaired wound healing with several pathogenic phenotypes, and literature indicating that STZ treatment is one of the most reliable approaches to generate impaired healing in rodents^{44,45,51}, we believe our model sufficiently captures key aspects of pathological wound repair relevant to clinical translation. Moreover, our previous and newly added findings, subjected in new Fig. 8, clearly demonstrate the efficacy of partial epidermal reprogramming in this disease model to enhance wound repair, further supporting the clinical applicability of our findings.

Specific points:

Q3. Line 97: K14-CreER mice do not induce specific deletion in epidermal keratinocytes. The K14 promoter is also active in hair follicle keratinocytes and basal epithelial cells of other stratified epithelia. This sentence should be modified. In particular, it is unclear if and to what extent reprogrammed hair follicle keratinocytes contribute to the observed effects.

Thank you for the suggestion and comments. We agree that Krt14 activity is not restricted solely to the interfollicular epidermis, as it is also expressed in hair follicles and some other stratified squamous epithelia. Accordingly, we revised the sentence to remove the word “specific”.

Although the Krt14 promoter does not induce an entirely specific induction, both our observations and previous reports indicate that the relative efficiency of gene induction is substantially higher in the interfollicular epidermis than in hair follicles when using Krt14-driven CreER systems. In our model with the administered tamoxifen (TAM) and DOX dosages, we consistently observed that TAM-induced GFP expression, and the subsequent mCherry expression, were much more robust in keratinocytes of the interfollicular epidermis compared to those in the hair follicles (new Supplementary Fig. 1b,c). Consistently, transient OSKM expressions induced under the Krt14 promoter did not alter the proliferative capacity of hair follicle cells (Rebuttal Fig. 2), unlike the interfollicular epidermis, where cell proliferation was markedly increased (Fig. 1j,k).

Fluorescence induction in hair follicles of Epi-iOSKM mice was detected only at very low levels compared to the interfollicular epidermis (new Supplementary Fig. 1b,c)

No detectable changes were observed in hair follicles of Epi-iOSKM mice (Rebuttal Fig. 2)

Despite the low efficiency of OSKM-mCherry expression in hair follicular keratinocytes, we

examined whether hair follicle cells influence the phenotypes observed in the Epi-iOSKM skin. As expected, the expression of EGF ligands (*Tgfa*, *Epgn*, and *Areg*), which can induce non-cell-autonomous effects in mCh^{neg} interfollicular epidermal keratinocytes, as well as immune cytokines (*Tnf*, *Ccl8*, *Cxcl10*, *Ccl20*, *Il1b*, and *Cxcl16*), which can induce T cell recruitment and activation, were not significantly altered in Epi-iOSKM hair follicle keratinocytes (Rebuttal Fig. 3). These findings suggest that hair follicles likely had little influence on the phenotypes we reported in our study.

Hair follicles of Epi-iOSKM mice show no detectable alterations (Rebuttal Fig. 3)

Consistently, K14-CreER mice under control of the human K14 promoter have been widely used in studies specifically examining the characteristics of the IFE cells²⁶⁻²⁸. Moreover, Zhang et al. (*Cell Cycle*, 2010) reported that even among several Krt14-CreER strains, the line used in our study (Jax #005107) exhibits relatively low sensitivity to tamoxifen and highly inefficient recombination within hair follicles. They recommended another mouse called K14-CreER2 (Jax #038390) as a better model for studying hair follicle stem cells²⁵. Additionally, because we induced gene expression from K14-CreER using the Dox-inducible system, the observed activity may have been concentrated in the more efficient interfollicular epidermis. Taken together, while some recombination in hair follicle basal cells is expected, the K14 promoter functions most strongly and consistently in the epidermis. Thus, in the context of our study, the induced genetic changes predominantly reflect biology in the epidermis rather than in hair follicles.

Inefficient recombination was observed within hair follicles when using the same K14-CreER driver in our system (Zhang et al., *Cell Cycle*, 2010)²⁵

Q4. The K14-CreER mice are known to be “leaky”. The authors should include GFP and mCherry fluorescence pictures of mice that were not treated with Dox.

Thank you for this question, which allows us to include GFP and mCherry fluorescence data without Tam/Dox treatment. We have newly added these data in Supplementary Fig. 1d, confirming chemically induced target gene expression.

K14-CreER mice may exhibit a very low background level of CreER activity, as has been

occasionally reported⁵². However, in our experiments, no GFP expression was observed prior to Tam induction, and no mCherry expression was detected prior to Dox treatment (new Supplementary Fig. 1d). Consistently, previous studies using this K14-CreER line (Jax #005107) also reported the absence of leakiness in the absence of tamoxifen⁵³. Thus, any leaky activity was negligible in our system and did not affect the reliability or specificity of our results.

No detectable leakiness in the absence of tamoxifen and doxycycline in our system (new Supplementary Fig. 1d)

Q5. In Fig. 1b the authors should add a high magnification hematoxylin/eosin staining of the skin. This is important to judge the overall histological phenotype of the mice.

Thank you for the comments and suggestions. In addition to the results in Fig. 1b, we carefully examined whether short-term OSKM expression induces histological alterations in the skin using H&E staining. Our data show that three days of OSKM induction in epidermal keratinocytes did not cause detectable abnormalities in overall skin or epidermal architecture (new Supplementary Fig. 1e). Consistently, the epidermal thickness of Epi-iOSKM mice was comparable to that of control mice (new Supplementary Fig. 1f).

Representative high-magnification H&E staining images of Epi-iOSKM skin (new Supplementary Fig. 1e,f)

Q6. It is not clear what percentage of basal and suprabasal keratinocytes express OSKM. This information should be provided.

Thank you for the comments. We obtained reliable quantitative data on the mosaic ratio in both the basal and suprabasal compartments of the Epi-iOSKM epidermis. Overall, the proportion of Sox2⁺ cells in the Epi-iOSKM epidermis was 23.10 ± 3.39%. Further quantitative analysis showed that Sox2⁺ cells accounted for 21.91 ± 3.97% in the Krt14⁺ basal layer and 25.55 ± 3.43% in the Krt14⁻ suprabasal layer, indicating that OSKM induction efficiency was comparable between basal and suprabasal keratinocytes. This new data is provided in the new Supplementary Fig. 1j.

Proportion of OSKM-induced cells in basal and suprabasal compartments of the Epi-iOSKM epidermis (new Supplementary Fig. 1j)

Q7. Expression of *Klf4*, *c-Myc* and *Oct-4* in the epidermis should also be confirmed (at least at the RNA level). This information should be available from the scRNA-seq data.

Thank you for this comment. Our system incorporates the TetO-OSKM-mCherry transgene (vector map provided below), in which mCherry expression reliably marks OSKM induction, consistent with previous reports⁵⁴.

In the original manuscript, we verified co-expression of Sox2 and mCherry for the induction of TetO-OSKM-mCherry transgene (Fig. 1c). To further strengthen this point, we performed new immunostaining analyses and confirmed the induction of Oct4, *Klf4*, and *c-Myc* in mCh^{pos} cells of the Epi-iOSKM epidermis upon Dox treatment. These new data are included in new Supplementary Fig. 1g-i of the revised manuscript.

Immunostaining showing Oct4 expression in mCh^{pos} cells (new Supplementary Fig. 1g)

Immunostaining showing *Klf4* expression in mCh^{pos} cells (new Supplementary Fig. 1h)

Immunostaining showing *c-Myc* expression in mCh^{pos} cells (new Supplementary Fig. 1i)

Q8. The scRNA-seq data suggest alterations in apoptosis. Does reprogramming affect

apoptosis of mCherry+ and mCherry- keratinocytes?

Thank you for this comment. To examine whether partial reprogramming induces apoptosis in mCh^{pos} or mCh^{neg} keratinocytes, we performed immunostaining for cleaved caspase-3. No significant signal was detected, indicating that partial epidermal reprogramming does not induce apoptosis in the Epi-iOSKM epidermis. These data have been added as the new Supplementary Fig. 2c.

Immunostaining for cleaved caspase-3 revealed no detectable apoptosis upon partial epidermal reprogramming (new Supplementary Fig. 2c)

We further investigated why the ontology term “programmed cell death” was changed upon partial reprogramming (Fig. 2a; Supplementary Fig. 5c) by analyzing the leading-edge differentially expressed genes (DEGs) that contributed most to this enrichment. Genes related to inflammation and epidermal differentiation, including *Ntrk1*, *Cxcl10*, *Ccl2*, *Slc40a1*, and *Il1b*^{27,55-58}, were upregulated. This suggests that cell death itself is not induced; rather, apoptosis-associated genes—many of which are also linked to regenerative processes—are increased. Another gene, *Tmigd1*, one of the most critical contributors to the “programmed cell death” enrichment, was upregulated upon partial epidermal reprogramming; however, this gene is known to inhibit apoptosis^{59,60}, further supporting our cleaved caspase-3 staining data showing that partial reprogramming does not promote apoptotic cell death in either mCh^{pos} or mCh^{neg} keratinocytes.

Q9. The reversibility of the reprogramming effect is particularly important. Are the mCherry+ cells completely lost at day 10 after Dox withdrawal? If yes, do they undergo apoptosis or are they lost by terminal differentiation?

Thank you for the comments. Our mouse model utilizes a Dox-inducible gene expression system (Fig. 1a), in which *Oct4*, *Sox2*, *Klf4*, *Myc*, and *mCherry* are expressed only during the period of Dox administration⁵⁴ (please see our response to question #4 and #7). In this Tet-On system, doxycycline binds to rtTA, inducing a conformational change that increases its affinity for the tetO binding site, as previously described⁶¹. This activation was validated in our model by the expression of mCherry and OSKM (Fig. 1b,c; new Supplementary Fig. 1d,g-j). Upon Dox withdrawal, ectopic gene expression ceases. Indeed, *mCherry* signals were almost absent three days after Dox withdrawal (Supplementary Fig. 2b), and no cell death was detected (new Supplementary Fig. 2c,d). Consequently, in our system, *mCherry*+ cells were not observed 10 days after Dox withdrawal; however, it cannot be determined whether the cells that expressed OSKM-mCherry during Dox treatment remained or were lost by that time.

(Fig. 1a)

Loss of mCherry signal after Dox withdrawal without cell death (new Supplementary Fig. 2b,d)

Our data support the conclusion that partially reprogrammed cells undergo normal differentiation within the epidermal lineage. They acquired regenerative cellular characteristics (Fig. 2) and enabled them to return to the normal epidermal differentiation program. This is likely because three days of Dox treatment endowed the cells with plasticity and regenerative characteristics without causing escape from the epidermal lineage. Consistently, after partial epidermal reprogramming, none of the cells expressed the full reprogramming marker *Nanog*, whereas most of the cells expressed epidermal cell lineage markers such as *Krt5* and *Trp63* (Supplementary Fig. 7i-k).

Partial epidermal reprogramming does not drive cells to escape from the epidermal lineage (Supplementary Fig. 7i-k)

Partial reprogramming confers epidermal lineage-preserved plasticity (Fig. 2n)

Fig. 2n illustrates our partial reprogramming model, in which cellular plasticity is conferred within epidermal cell lineages. IFE cells that acquired plasticity through partial reprogramming were converted into a state resembling IFE cells observed during a natural wound healing situation (Fig. 2). In typical wound healing, such wound-responsive cells return to the original state of normal IFE cells after healing^{62,63}, and the partially reprogrammed cells in our study likely followed a similar trajectory.

Consistently, after 10 days of Dox withdrawal, the Epi-iOSKM epidermis had returned to its normal state. The induction of *Krt17/6a* was reversed (Supplementary Fig. 7f,g), demonstrating that the effects of partial reprogramming are reversible. Most cells expressed *Krt14* or *Krt10*, markers of basal and suprabasal epidermis, respectively, and the proportions of cells expressing these markers were comparable to those in normal epidermis (Supplementary Fig. 7h).

This recovery occurred without cell death through apoptosis: Immunostaining for cleaved caspase-3 revealed no detectable signal in the Epi-iOSKM epidermis during Dox treatment (Supplementary Fig. 2c; see also our response to reviewer 2, question #8) or after Dox

withdrawal (Supplementary Fig. 2d; Rebuttal Fig. 5). These findings suggest that the Epi-iOSKM epidermis returned to its original state without apoptotic cell death. Collectively, the partially reprogrammed cells likely re-acquired normal epidermal cell identity. Considering that the average cycle for mouse epidermal stem cells to fully differentiate is approximately 10 days^{64,65}, by 10 days after discontinuation of Dox, it can be expected that some cells were likely differentiating within the normal epidermal cell lineages, while others had already been shed through terminal differentiation. Taken together, the complete reversibility of the changes highlights the potential safety of partial epidermal reprogramming for therapeutic applications, as it avoids permanent alteration of cell identity.

Q10. Fig. 3g and h: PI3K inhibition is likely to affect epidermal cell proliferation in general. The authors should add control mice treated with the PI3K inhibitor for comparison. Same for EGFR inhibitor in Fig. 4.

Thank you for the comments, which allowed us to strengthen our data by comparing the effects of PI3K and EGFR inhibition on Epi-iOSKM epidermis with those on control epidermis. We revised Figs. 3h and 4h, which originally showed the reversal of *Krt17* expression following PI3K or EGFR inhibitor treatment, by adding new control groups in which the inhibitors were applied to the skin of control mice. These results demonstrated that PI3K and EGFR inhibition reduced the *Krt17* level in the Epi-iOSKM epidermis to levels similar to those in the control epidermis.

(revised Figs. 3h and 4h)

In control epidermis, the inhibitor effects were not significant. As the reviewer noted, PI3K and EGFR pathways could influence *Krt17* expression in control epidermis, either directly or indirectly, for example, by regulating cell proliferation^{9,66}. However, as shown in new Figs. 3h and 4h, endogenous *Krt17* levels are already low in normal epidermis. Moreover, PI3K and EGFR activity is also low in normal epidermis, as shown in Figs. 3f and 4f. Consistent with this, previous studies also reported that baseline p-EGFR and p-AKT signals are generally low in the epidermis and often near the detection limit⁶⁷⁻⁶⁹. Therefore, inhibitor treatments in control epidermis, which already exhibited low *Krt17* expression and low PI3K/EGFR activity, would not have dramatic effects, as observed in our experiments. In conclusion, PI3K and EGFR pathways are essential for *Krt17* expression specifically in the Epi-iOSKM epidermis.

For Figs. 3g and 4g, which examined HIF-1 α activity, we also added data comparing the effects of inhibitor treatment on control epidermis in Supplementary Figs. 8c and 9e, respectively. Similar to the results for *Krt17*, inhibitors had no significant effects in control mice, confirming that PI3K and EGFR inhibition reduced HIF-1 α activity only in the Epi-iOSKM epidermis.

Q11. The result in Fig. 3g requires quantification.

Thank you for the comments. We have added the quantification data of Fig. 3g in new Supplementary Fig. 8c, which confirms the reduction observed upon PI3K inhibition. Please see also our response to question #10.

Q12. Fig. 3j: It has not been shown that PI3K directly regulates keratin 6/16/17. This may be simply a consequence of the increased proliferation.

Thank you for this comment. We agree that our data do not demonstrate a direct regulation of Krt6a/16/17 by PI3K. Accordingly, we have revised the model in Fig. 3j to avoid implying direct causality.

Functionally, PI3K inhibition with LY294002 reversed Krt17 and Krt6a upregulation in Epi-iOSKM epidermis (Fig. 3h; Supplementary Fig. 8d), indicating PI3K activation is essential for Krt17/6a induction upon partial epidermal reprogramming. Although we cannot determine whether PI3K regulates Krt17/6a directly or indirectly, our data demonstrate that OSKM expression in epidermal keratinocytes activates the PI3K pathway, and this activation led to Krt17/6a upregulation as depicted in Fig. 3j.

A point that was not fully elucidated in Fig. 3j is the relationship between the phenotypes observed in Epi-iOSKM epidermis, such as increased proliferation, Krt17/6a expression, and HIF-1α activation. Actually, these are closely related: HIF-1α activation can induce cell proliferation^{70,71}; Krt17 expression also can induce cell proliferation^{11,72}; Krt17 expression level is correlated with enhanced proliferation^{9,10}; and Krt17 expression level increases under hypoxia/HIF-1α activation conditions⁷³⁻⁷⁵. In addition, EGFR activation via autocrine effects can also result in HIF-1α and Krt17 induction. Collectively, the outcomes of partial

reprogramming, mirroring injury responses, can influence each other. To disentangle these effects, we utilized single cell transcriptomics, which distinguishes proliferating cells from non-proliferating basal/suprabasal cells, allowing us to discern correlations between increased proliferation and other phenomena such as Krt17/6a induction. Importantly, in Epi-iOSKM epidermis, our scRNA-seq data revealed that KRT17/6a was upregulated in non-proliferating Basal and Supra mCh^{pos} cells (Fig. 2g). This suggests that the PI3K pathway may contribute to Krt17/6a induction in two ways: (1) by expanding the pool of proliferating cells that intrinsically express high levels of Krt17, and (2) by elevating Krt17/6a expression even in non-proliferating basal and suprabasal cells. This highlights that the observed induction cannot be explained solely by enhanced proliferation.

In summary, while our data demonstrate that Krt17 induction in Epi-iOSKM epidermis is not simply a consequence of increased proliferation, the complex correlations among proliferation, HIF-1 α , and keratin expression cannot be fully disentangled at present. To avoid misinterpretation, we revised Fig. 3j so that the updated illustration no longer implies a direct link, thereby minimizing potential confusion.

Q13. Line 292: NRG (neuregulin) does not activate the EGFR – it activates ERBB3 and ERBB4.

Thank you for the comments. We agree with the reviewer that neuregulins (NRGs) are ligands for ERBB3 and ERBB4, but not direct ligands for EGFR (ERBB1)⁷⁶. However, ERBB3 lacks intrinsic kinase activity and requires heterodimerization with other ERBB family members, such as EGFR or ERBB2, to transduce signals⁷⁷. Indeed, EGFR/ERBB3 heterodimers can be activated in a complementary manner, with EGF binding to EGFR and NRG binding to ERBB3, thereby inducing full complex activation^{78,79}. In this context, the upregulation of NRG observed in our data likely contributes to EGFR/ERBB3 complex activation, rather than direct EGFR activation.

We have revised the text to clarify this point as follows:

“Notably, signaling pathways associated with the ECM, such as FN1 and KLK, those activating EGFR, such as EPGN and EGF, and those promoting EGFR/ERBB3 complex signaling, such as NRG, were significantly activated in the Epi-iOSKM epidermis.”

Q14. Fig. 4h: Please confirm the efficient EGFR inhibition by immunofluorescence staining for pEGFR.

Thank you for the comments. We have added immunofluorescence staining data demonstrating effective inhibition of EGFR phosphorylation by AG1478, now provided in new Supplementary Fig. 9d.

Effective inhibition of EGFR activation in Epi-iOSKM epidermis by AG1478 (new Supplementary Fig. 9d)

Q15. The model proposed in Fig. 4j is interesting, but it is only based on immunofluorescence stainings. It is important to test if PI3K activation in cultured keratinocytes promotes EGFR activation.

Thank you for the comments. We performed *in vitro* experiments using HaCaT human keratinocyte cell lines. Treatment with the PI3K activator 740 Y-P for 24 hours increased phospho-AKT levels, a hallmark for PI3K pathway downstream activation. Interestingly, the phospho-EGFR level was elevated together. We added this data as new Supplementary Fig. 9f.

It is known that EGFR can function as upstream of the PI3K pathway. However, PI3K reciprocally affects EGFR activity, with context-dependent outcomes. Among several related reports, there are also some studies showing that PI3K activation induces EGFR activation, consistently with our results: for example, PI3K-AKT activation is required for UVB-induced EGFR activation in human keratinocytes⁸⁰; PIK3CA gain-of-function mutations induce AREG-EGFR-ERK signaling⁸¹; and PI3K causes reactive oxygen species^{82,83}, which can activate EGFR phosphorylation in keratinocytes⁸⁴.

Collectively, our original data and the new data prompted by this comment, emphasize that PI3K activation is a crucial intermediate step in partial epidermal reprogramming.

PI3K activation induces EGFR activation in HaCaT cells (new Supplementary Fig. 9f)

Q16. Line 321: IL1 and VEGF do not directly regulate keratinocyte proliferation. This sentence should be modified.

Thank you for the comments. While IL-1 may not have direct mitogenic effects on keratinocytes, *in vivo*, it can promote keratinocyte proliferation through indirect pathways⁸⁵. For example, IL-1 β activation in epidermal keratinocytes has been reported to induce hyperplasia⁸⁶. Although IL-1 β itself does not act directly, it can indirectly stimulate keratinocyte proliferation by activating immune cells and fibroblasts^{87,88}.

VEGF has also been reported to promote keratinocyte proliferation. VEGF signaling via keratinocyte-expressed VEGFR1 can enhance keratinocyte proliferation⁸⁹. In addition, correlations between VEGF and/or VEGFR1 expression and increased proliferation have been reported in wound healing and inflammatory pathological processes^{90,91}.

To avoid over-interpretation, we have revised the sentence in the manuscript as follows:
“The cellular pathways identified in the Epi-iOSKM epidermis were associated with increased

proliferation; EGF signaling, a key modulator for keratinocyte proliferation, as well as IL1 and VEGF signaling that can indirectly contribute, were significantly enriched.”

Q17. SFRP1 is a secreted protein, which should act on mCherry- and on mCherry+ cells. How do the authors explain a selective effect on mCherry- cells if SFRP1 is indeed responsible?

Thank you for the comments. We proposed that *Sfrp1* overexpression and *Wnt4/6/16* reduction contributed to WNT pathway downregulation in mCh^{neg} cells of Epi-iOSKM epidermis. As the reviewer noted, secreted *Sfrp1* could affect both mCh^{pos} and mCh^{neg} cells; therefore, the observed differences suggest the involvement of an additional regulator.

We identified PI3K pathway activation specifically in mCh^{pos} cells (Fig. 3e,f). This can activate the WNT-β-catenin pathway by inhibiting the β-catenin repressor GSK-3β^{92,93}. Thus, mCh^{pos} cells are simultaneously affected by activation of PI3K pathway and gene expression changes to inhibit WNT ligands activity, respectively. In contrast, mCh^{neg} cells are affected only by gene expression changes, such as *Wnt4*, *Wnt6*, *Wnt16* and *RSPO3* downregulation and *Sfrp1* upregulation, which can downregulate WNT-β-catenin pathway (Supplementary Fig. 10b,c). Accordingly, WNT-β-catenin pathway downregulation was observed exclusively in mCh^{neg} cells of Epi-iOSKM epidermis (Supplementary Fig. 10a).

We have discussed this model, which may illustrate the mechanisms underlying cellular pathway alterations in the Epi-iOSKM epidermis in the revised manuscript as follows:

“Despite these changes in gene expression suppressing the WNT pathway, we observed that the activity of WNT pathway was not altered in mCh^{pos} cells (Supplementary Fig. 10a). This may be the involvement of mCh^{pos} cells-specific PI3K pathway activation. PI3K-AKT activation is known to activate WNT-β-catenin pathway by inhibiting GSK-3β, a repressor of β-catenin.”

Proposed model illustrating mechanisms underlying WNT pathway activity regulation in mCh^{pos} and mCh^{neg} cells (Rebuttal Fig. 6)

Q18. Supplementary Fig. 5e: An important control is missing: LGK alone (same for the migration experiment). It seems likely that this compound already inhibits cell proliferation in control cells. The very weak effect of EGF on proliferation is surprising.

Thank you for the comments. We have now included LGK alone control in our *in vitro* experiments. The revised data are provided in Supplementary Fig. 10e,f (original Supplementary Fig. 5e,f).

The weak effect of EGF observed in our earlier results is likely due to the inherently high proliferative capacity of the cell lines and the presence of growth factors in the culture media. To address this, we performed the proliferation assay under starvation conditions, which revealed a significantly greater increase in proliferation upon EGF treatment.

***In vitro* EdU cell proliferation assay including LGK alone control (revised Supplementary Fig. 10e)**

***In vitro* scratch wound healing assay including LGK alone control (revised Supplementary Fig. 10f)**

Despite the well-known proliferative effects of the WNT pathway, the LGK alone group did not show a significant reduction in proliferation compared to control cell. This indicates that the inhibitory effect of LGK is particularly pronounced in keratinocytes whose proliferative ability is elevated by EGF. This aligns with prior studies showing that the effects of WNT inhibition depend on cell state or experimental conditions⁹⁴. Specifically, it is consistent with earlier reports that porcupine inhibitors including LGK974 suppress the excessive proliferation of oncogenic mutant cells and tumor growth without affecting the proliferation of normal adult stem cells^{95,96}.

Together, the added control supports that WNT pathway inhibition could more effectively counteract the promotion of proliferative capacity by EGFR activation in the Epi-iOSKM epidermis. This further demonstrates that mCherry-neg cells in the Epi-iOSKM epidermis, simultaneously affected by upregulated EGFR signaling and downregulated WNT signaling, acquire properties favorable for wound healing without increased proliferation.

Q19. Line 374: Epidermal T cells in mice are almost exclusively gd T cells. These cells are not recruited upon injury. They only populate the skin once during embryonic development (see publications from W. Havran and A. Hayday). The increase in these cells is likely a consequence of increased T cell proliferation, which should be tested. Alternatively, there may be an abnormal recruitment of ab T cells to the epidermis, which seems likely because there are more RORgt cells. Is there an increase in epidermal and dermal ab or gd T cells or both?

Thank you for the comments and suggestions. We have supplemented our data to clarify the origin of the increased T cells observed in the skin of Epi-iOSKM mice.

The increase in epidermal T cells upon partial epidermal reprogramming could result either

from the proliferation of resident epidermal T cells or from the migration of T cells from other sites. In the original manuscript, we showed that increased ectopic ROR γ t cells—normally absent in the steady-state epidermis—in Epi-iOSKM skin had migrated from elsewhere, but we didn't assess how epidermal T cell proliferation was affected.

To test this, we analyzed the proportion of Ki67-positive proliferating cells among epidermal T cells. Both scRNA-seq and immunohistochemistry analyses demonstrated that partial epidermal reprogramming does not alter epidermal T cell proliferation. In the scRNA-seq data, the proportions of Ki67-positive cells among T cells were 6.25% (6/96) in controls and 6.45% (14/217) in Epi-iOSKM epidermis, indicating comparable levels between the two conditions (new Supplementary Fig. 11c). Consistently, immunohistochemistry revealed no difference in the proportion of CD3 and Ki67 double-positive cells between control and Epi-iOSKM epidermis (new Supplementary Fig. 11d). These results indicate that the increased T cells in Epi-iOSKM epidermis result from migration rather than local proliferation. The related data have been added in the new Supplementary Fig. 11c,d.

T cell proliferation remains changed in Epi-iOSKM epidermis (new Supplementary Fig. 11c,d)

As the reviewer noted, most T cells in the murine epidermis are $\gamma\delta$ T cells—particularly, $V\gamma 3^+$ T cells—that originate during embryonic development⁹⁷. During injury repair, however, additional subsets of $\gamma\delta$ T cells are further recruited to the epidermis; most of these are $\gamma\delta$ T17 cells that express ROR γ t⁹⁸⁻¹⁰⁰. ROR γ t⁺ T cells can be $\alpha\beta$ Th17 cells or $\gamma\delta$ T17 cells, and among these, ROR γ t⁺ $\gamma\delta$ T17 cells are recruited to the epidermis and dermis via the CCL20-CCR6 axis¹⁰⁰⁻¹⁰² during wound healing. Consistently, partial epidermal reprogramming recruited ROR γ t⁺ $\gamma\delta$ T cells to the epidermis. It was demonstrated that partial epidermal reprogramming recruited ROR γ t⁺ cells by CCL20-CCR6 axis similarly with healing skin in the previous version of the manuscript (Fig. 5f). To further validate this, we examined the expression of marker genes for $\alpha\beta$ and $\gamma\delta$ T cells in our epidermal scRNA-seq data. *Trdc*, which encodes the delta chain of the T cell receptor and serves as a marker for $\gamma\delta$ T cells, was highly expressed at comparable levels in both control and Epi-iOSKM skin (Rebuttal Fig. 7). In contrast, *Trac*, which encodes the alpha chain of the T cell receptor and is a marker for $\alpha\beta$ T cells, was nearly absent in both groups. These results indicate that the infiltrating epidermal T cells are predominantly $\gamma\delta$ T cells (Rebuttal Fig. 7).

No increase in $\alpha\beta$ T cell populations in the Epi-iOSKM epidermis (Rebuttal Fig. 7)

We did not determine whether the recruited T cells in the dermis were $\alpha\beta$ or $\gamma\delta$ T cells, as this would require additional time and resources. Nevertheless, it is plausible that the dermis harbors similar T cell subsets to those observed in the epidermis, since CCL20–CCR6–mediated recruitment of ROR γ t⁺ cells likely occurs throughout the skin without a strict distinction between epidermal and dermal compartments (Fig. 5f, please see also response to reviewer 2, question #20).

Q20. Lines 390-392: The effect of the CCR6 antagonist on the number of epidermal ROR γ t cells is not significant.

Thank you for the comments. In the dermis, the increase in the number of ROR γ t[±] cells was completely abrogated by CCR6 antagonist treatment. In the previous manuscript (lines 390-392), our point was that although the effect of the antagonist was less pronounced in the epidermis than in the dermis, the significant difference in the number of ROR γ t⁺ cells between control and Epi-iOSKM epidermis was also eliminated under CCR6 antagonist treatment (Rebuttal Fig. 8). Thus, it cannot be excluded that the mechanisms via CCR6 also operate in the epidermis.

Previous Fig. 5f in the original manuscript (Rebuttal Fig. 8)

In the revised manuscript, we now provide newly quantified data that more reliably demonstrate the contribution of the CCL20-CCR6 axis (revised Fig. 5f). In normal skin, ROR γ t⁺ cells are absent in the epidermis and present only in small numbers in the dermis (the first image of revised Fig. 5f, left). Upon activation of the CCL20-CCR6 axis via epidermal CCL20 expression, ROR γ t⁺ cells migrate toward the epidermis, leading to an overall increase in ROR γ t⁺ cells in both epidermis and dermis, with more accumulation in the adjacent dermis¹⁰⁰⁻¹⁰². Because the CCL20-CCR6 axis recruits ROR γ t⁺ cells to both compartments through the same mechanism, the most accurate way to evaluate its function is to quantify cells across the whole skin, rather than separating epidermis and dermis. Consistently, our revised Fig. 5f, which quantifies total skin ROR γ t⁺ cells, shows a significant antagonist effect.

As ROR γ t⁺ cells are far more abundant in the dermis than in the epidermis (revised Fig 5f, left), this result is in line with our previous quantifications showing a clear antagonist effect in the dermis. In the original manuscript, we analyzed epidermis and dermis separately to align with total T cell quantification in Fig. 5c. However, due to the very low number of ROR γ t⁺ cells in the epidermis, such comparisons are less reliable. Thus, the integrated quantification presented here provides a more accurate assessment of CCL20-CCR6 axis function and supports its key role in recruiting ROR γ t⁺ cells, as described in the revised manuscript.

CCR6 antagonist rescues the recruitment of RORγt⁺ cells in the Epi-iOSKM skin (revised Fig. 5f)

Q21. Page 16, first paragraph: It is not surprising that no tumors were observed after 10 days of Dox treatment, because epidermal skin tumors develop slowly. It may well be that tumors would develop at a later time point (perhaps even after discontinuation of Dox treatment). This possibility should at least be discussed. In any case, it is clear from these results that long-term OSKM induction in keratinocytes is detrimental.

We appreciate this comment, as it prompted us to further examine the safety validation of our Epi-iOSKM model. Research on *in vivo* partial reprogramming, which inherently carries certain risks^{6,103}, requires rigorous safety validation. While previous studies have reported adverse outcomes from whole-body OSKM expression in mice^{104,105}, our results show that short-term OSKM induction in keratinocytes (3 days) did not cause any deleterious phenotypes. In line with the reviewer's suggestion, we have now added detailed histological analyses demonstrating the absence of histological abnormalities in this condition as new Supplementary Fig. 1e,f (please see also response to reviewer 2, question #5).

To further assess potential risks, we extended Dox treatment to 9 days and observed inflammatory phenotypes (Fig. 5h-j). Notably, after 10 days of Dox withdrawal, these inflammatory phenotypes persisted. The results of this experiment were described in the revised manuscript as follows:

“Moreover, following 10 days of Dox withdrawal after a 9-day administration period, the mice exhibited other pathologies, such as dry skin, scaly ears, and hair loss, along with abnormal skin histology characterized by excessive thickening of the stratum corneum and an increased number of CD3⁺ T cells in both the epidermis and dermis (Supplementary Fig. 12g-i).”

Skin inflammation pathologies induced by 9-day prolonged OSKM expression in the epidermis persisted for 10 days (Supplementary Fig. 12g-i)

To address the reviewer's comments regarding whether prolonged OSKM induction in keratinocytes for 9 days might lead to cancer or other severe long-term outcomes after a long time has passed, we examined Epi-iOSKM skin phenotypes 30 days after DOX withdrawal following 9 days of OSKM induction. Unexpectedly, our new findings suggested the reversibility of the induced phenotypes by 9-day induction.

The epidermis recovered from hyperplasia (new Supplementary Fig. 12j,k) and the pluripotency marker Nanog was not expressed (new Supplementary Fig. 12l). These findings indicate that prolonged OSKM expression for 9 days did not cause progressive epidermal

pathology over time. We speculate that this reversibility may be due to the cells that expressed OSKM did not reach a pluripotent state. The phenotypes observed after 9 days of OSKM induction were consistent with inflammatory changes associated with epidermal hyperplasia (Fig. 5h-j), rather than neoplasm formation (Supplementary Fig. 12b,c). In many instances, inflammation caused by epithelial perturbations without systemic immune compromise can be reversible within weeks to months, even when initially severe¹⁰⁶. Our findings suggest that the Epi-iOSKM phenotype may follow a similar, reversible situation.

Recovery from the prolonged OSKM expression effects (new Supplementary Fig. 12j-l)

Although partial epidermal reprogramming is not fatal, unlike *in vivo* partial reprogramming in other tissues, it is evident that the longer continuous OSKM induction will eventually lead to deleterious effects. Therefore, we assessed the period of time required for such fatal outcomes, including tumorigenesis or death. In other tissues, OSKM induction caused death within only a few days^{103,105}; however, even after long-term partial reprogramming of epidermal keratinocytes for a month, none of the six treated mice died.

We further extended the Dox treatment durations. When the Dox treatment period lasted for more than six weeks, mice began to die. Using the surviving mice, we examined the results of 60 days of OSKM expression and observed the formation of tumors. Cells in the tumor highly expressed Nanog, suggesting teratoma formation (Rebuttal Fig. 1).

Nanog induction and teratoma formation after 2 months of OSKM expression (Rebuttal Fig. 1)

In summary, *in vivo* OSKM expression in epidermal keratinocytes for three days was beneficial for wound healing without adverse effects; for nine days, it induced skin inflammation that is recovered over time; and for two months, it caused both teratoma formation and mortality.

Q22. The effect of partial reprogramming on wound closure is minor (Supplementary Figure S7A). This is not too surprising, because wound healing in healthy mice is highly optimized. The data on the length of the epithelial tongue and the transepidermal water loss are more convincing. It is essential to show histological stainings of cross-sections from wounds at

different time points. Is the area of granulation tissue also affected?

Thank you for the comments and suggestions. In our original analysis, we focused on the analysis of repairing wound-edge sections, consistent with recent notable studies in this field^{12,98}, enabling a detailed assessment of cell- and tissue-level dynamics with diverse markers. We have now added cross-sectional images as requested, which, together with our original analysis, provide a broader characterization of the healing process. In particular, we assessed granulation tissue formation from H&E images of whole-wound cross-sections at both early (3 dpw) and middle (10 dpw) stages of wound healing. We added these data as new Supplementary Fig. 13b,c.

Because we employed a splinted wound model and covered it with a semi-occlusive transparent film to maximize the effects of re-epithelialization, full-thickness wounds heal without scab formation, as previously reported¹⁰⁷. In this model, granulation tissue thickness was increased in Epi-iOSKM mice compared to controls at both 3 and 10 dpw (new Supplementary Fig. 13b,c). In the original manuscript, we also reported that the scar thickness reduced after wound closure (Fig. 6d), and collagen composition was more rapidly recovered in Epi-iOSKM mice (Fig. 6e,f; Supplementary Fig. 13e-i). The new and previous results show that granulation tissue formation is promoted during healing, and the tissue rapidly returns to its original state after healing, respectively, further supporting that partial epidermal reprogramming enhances dermal healing.

Increased granulation tissue formation in wounded Epi-iOSKM skin (new Supplementary Fig. 13b,c)

Q23. It is unclear if mCherry+ or mCherry- keratinocytes preferentially contribute to re-epithelialization. This is mechanistically important and should be checked at different time points.

Thank you for the comments and suggestions. It would be valuable to apply a lineage-tracing tool to directly determine how mCh^{pos} and mCh^{neg} cells contribute to re-epithelialization, which appears to be the basis of the reviewer's suggestion. However, our current system does not allow such analysis. In our model, mCherry expression is limited to the period of Dox administration and ceases upon withdrawal (please see also response to reviewer 2, question #9). Consequently, after stopping Dox induction at 1 dpw, mCherry fluorescence cannot distinguish partially reprogrammed cells (mCh^{pos}) from their neighbors (mCh^{neg}), and indeed no mCherry signals were detected at 7 dpw. Although GFP fluorescence was observed in a substantial portion of the newly regenerative epithelium, this GFP⁺ population includes both mCh^{pos} and mCh^{neg} cells, indicating only that both contribute to re-epithelialization (Rebuttal Fig. 9).

Immunofluorescence staining of GFP and mCherry in migrating epithelial tongues (Rebuttal Fig. 9)

Despite this technical limitation, we confirmed that both mCh^{pos} and mCh^{neg} cells contribute to re-epithelialization, without an apparent preferential role, as evidenced by the key mechanisms underlying accelerated healing.

In unwounded skin, the most prominent difference between the two populations in the Epi-iOSKM epidermis was the presence of increased proliferation in mCh^{pos} cells (Fig. 1j,k; Supplementary Fig. 4i), while other phenotypes associated with regenerative characteristics were observed in both populations (mCh^{pos} and mCh^{neg} cells) (Fig. 2). We then examined epithelial cell proliferation during repair at 3 dpw, but found no significant differences between control and Epi-iOSKM skin (new Supplementary Fig. 14b), indicating that mCh^{pos} cell-specific proliferation during DOX treatment alone cannot account for accelerated re-epithelialization.

No significant difference in epidermal cell proliferation were observed between control and Epi-iOSKM skin at 3 dpw (new Supplementary Fig. 14b)

In the original manuscript, we found that HIF-1 α activity is increased in the healing epithelium even after discontinuation of Dox treatment (Fig. 7b,c), suggesting that sustained HIF-1 α activation could be a potential driver of enhanced re-epithelialization. In additional experiments, we confirmed that elevated HIF-1 α activity is required for the accelerated re-epithelialization observed in Epi-iOSKM skin. We have added this data as new Fig. 7i-k and new Supplementary Fig. 15a,b. The relevant details described in the revised manuscript are as follows:

“When a wound occurs in Epi-iOSKM mice, HIF-1 α is activated twice in the epidermis—first by partial reprogramming and again by the wound itself. To disrupt the synergistic effect of these two stimuli, we transiently inhibited HIF-1 α activity by treating the HIF-1 α inhibitor PX-478 once at the time of wounding (Fig. 7i). HIF-1 α activity decreased at 1 dpw but recovered by 10 dpw as expected (Supplementary Fig. 15a,b). Interestingly, transient HIF-1 α inhibition during the early repair phase reversed the accelerated re-epithelialization in Epi-iOSKM mice, bringing it to levels comparable to controls (Fig. 7j,k).”

Experimental scheme for HIF-1 α inhibition with wounding (new Fig. 7i)

Effective inhibition of HIF-1 α by PX-478 at 1 dpw with recovery at 10 dpw (new Supplementary Fig. 15a,b)

HIF-1 α activation is necessary for the enhanced re-epithelialization in Epi-iOSKM mice (new Fig. 7j,k)

Increased activation of HIF-1 α , essential for promoting re-epithelialization, was observed at comparable levels in both mCh^{pos} and mCh^{neg} cells of Epi-iOSKM epidermis (Fig. 2l and 7b), indicating that both cell populations contribute similarly to the accelerated re-epithelialization.

Q24. The analysis of immune cells is restricted to T cells. However, macrophages are much more important players in wound healing. Is there a difference in the number of macrophages?

Thank you for the helpful comments and suggestions. We assessed the macrophage proportions at the early stages of wound healing (3 and 7 dpw) and found no significant differences between control and Epi-iOSKM skin. These results have been included in the revised manuscript as new Supplementary Fig. 14e.

No significant difference in the number of macrophages was observed between control and Epi-iOSKM skin at 3 and 7 dpw (new Supplementary Fig. 14e)

As the reviewer noted, macrophages are important players in wound repair. They are

usually activated by recognizing microenvironmental changes immediately after injury, for example through pattern recognition receptors¹⁰⁸. Once activated, macrophages orchestrate various healing steps, including proliferation and migration of epithelial cells^{109,110}. In contrast, T cells are primarily recruited and activated under the influence of epithelial cells and myeloid cells following the inflammatory phase^{99,111}. Thus, it is reasonable that partial reprogramming of epidermal cells would have a greater effect on T cells than on macrophages.

Consistent with this, our data show that most immune cytokines significantly upregulated by partial epidermal reprogramming, including *Ccl20*, *Ccl8*, *Cxcl10*, and *Cxcl16* (Fig. 5e), are chemokines that primarily recruit T cells rather than macrophages^{99,112-114}. Taken together, these findings suggest that T cells, rather than macrophages, are the main mediators shaping the immune niche in Epi-iOSKM skin.

Q25. Line 468: Fig. 6i,j and Supplementary Fig. 7a do not show that wound healing is delayed in the streptozotocin-treated mice. This should be shown in one figure. Again, histological cross-sections from the wounds at different stages should be shown. This is important to fully judge the phenotypic alterations.

Thank you for the suggestions. By reorganizing the experimental control group setup, we enabled simultaneous examination of the inhibitory effects caused by STZ and the recovery effects induced by partial epidermal reprogramming within one figure.

Improvement of diabetic wound healing through partial epidermal reprogramming (new Fig. 8)

Wound closure data are provided in the revised Fig. 8b, and re-epithelialization data are provided in the revised Fig. 8d,e, each encompassing the three experimental conditions (normal control mice, diabetic control mice, diabetic Epi-iOSKM mice). These results clearly demonstrate that wound healing is delayed in STZ-treated disease models. Furthermore, we found reduced epidermal cell proliferation and angiogenesis during the diabetic healing process (new Fig. 8c,d,f), indicating that this model represents not merely delayed healing, but a disease model that disrupts multiple repair processes. Moreover, our data show that these defective phenotypes, both previously recognized and newly identified, were recovered through partial epidermal reprogramming (please see also responses to Reviewer 2, question #2).

We also provide H&E staining images of wound cross-sections at 3 dpw and 11 dpw with the same configuration as the data above: normal control mice, diabetic control mice, and diabetic Epi-iOSKM mice. This enabled an assessment of the overall histological alterations. Especially, the granulation tissue thickness was decreased in STZ-treated models than wildtypes as previously reported^{49,50} at 11 dpw, and this was restored by partial epidermal reprogramming (new Supplementary Fig. 16c,d).

Recovery of granulation tissue formation by partial epidermal reprogramming in diabetic impaired wounds (new Supplementary Fig. 16c,d)

Taken together, our reorganized and newly provided data clearly demonstrate that STZ treatment impairs multiple aspects of wound repair, and that partial epidermal reprogramming can effectively rescue these defects.

Q26. Line494: It is correct that some reduction in the extent of angiogenesis can reduce scarring in mice. However, this is an indirect effects, which is not very strong. The extent of scarring strongly depends on myofibroblast differentiation. The authors should check if this is affected in the OSKM mice. It is mechanistically unclear how OSKM expression suppresses scarring. It may also be a consequence of the enhanced reepithelialization.

Thank you for the comments, which give us the opportunity to further examine the mechanisms underlying reduced scarring in Epi-iOSKM mice. Through the analyses, we identified HIF-1 α as a key regulator modulating the enhancement of both epidermal and dermal repair in Epi-iOSKM mice.

As the reviewer noted, myofibroblasts can be contributors to regulating scar formation after wounding. To assess their presence, we performed immunostaining for α SMA, most commonly used marker of myofibroblasts¹¹⁵, in healing dermal region of control and Epi-iOSKM skin at 10 dpw. However, we did not detect significant differences in α SMA-positive myofibroblast abundance between two groups (new Supplementary Fig. 14d).

No significant differences in the number of α SMA⁺ myofibroblasts were observed between control and Epi-iOSKM skin at 10 dpw (new Supplementary Fig. 14d)

These results suggest that other mechanisms are involved. We therefore focused on HIF-1 α as a potential candidate. The rationale for focusing on HIF-1 α is described in detail in the revised manuscript as follows:

“Previous studies have reported that forced activation of HIF-1 α accelerates re-epithelialization. In our model, HIF-1 α activation induced by partial reprogramming was sustained after wounding (Fig. 7c), suggesting it may continue to promote re-epithelialization even after transient reprogramming has ceased. Moreover, the altered patterns of HIF-1 α activation correlated with reorganized angiogenesis (Fig. 7b-h), potentially influencing dermal repair. Notably, appropriate regulation of neovascularization during wound repair is known to reduce scar formation.”

Indeed, HIF-1 α activation in the epidermis was required for the reduced scarring in Epi-iOSKM mice. To test this, we transiently inhibited HIF-1 α during the early phase of wound healing (new Fig. 7i), which reversed the accelerated re-epithelialization observed in Epi-iOSKM skin (new Fig. 7j,k; see also responses to reviewer 2, question #23).

Experimental scheme for HIF-1 α inhibition with wounding (new Fig. 7i)

In addition to re-epithelialization, angiogenesis was also affected by HIF-1 α inhibition. We have included these new data in Fig. 7j,l and Supplementary Fig. 15c,d of the revised manuscript with the corresponding description as follows:

“Moreover, temporal inhibition of HIF-1 α markedly reduced angiogenesis. Without inhibition, control mice displayed widespread angiogenesis across the wound bed, whereas Epi-iOSKM mice showed angiogenesis concentrated near the original wound site (Fig. 7f-h). Following PX-478 treatment, however, both groups exhibited a pronounced reduction in angiogenesis, with low CD31 expression at newly formed dermis near both the initial wound sites and leading edges (Fig. 7j,l; Supplementary Fig. 15c,d). These findings indicate that HIF-1 α is a key regulator of neovascularization patterns during wound repair in both control and Epi-iOSKM mice.”

Disturbed angiogenesis by HIF-1 α inhibition during wound healing (new Fig. 7j,l; new Supplementary Fig. 15c,d)

Finally, we found that the scarring itself was also influenced by HIF-1 α , as shown in new Fig. 7m,n; new Supplementary Fig. 15e:

“We next examined whether HIF-1 α , which is essential for the alterations in re-epithelization and angiogenesis in Epi-iOSKM mice, also contributes to scar reduction. At 20 dpw, dermal thickness in the scarred area of PX-478 treated Epi-iOSKM mice was significantly thicker than that of untreated Epi-iOSKM mice and was comparable to that of PX-478 treated control mice (Fig. 7m). Consistently, the alterations in collagen I deposition observed in Epi-iOSKM skin were restored by PX-478 treatment (Supplementary Fig. 15e). In addition, the reduced proportion of collagen III⁺ area observed in Epi-iOSKM mice was reversed by HIF-1 α inhibition during partial epidermal reprogramming (Fig. 7n), indicating that elevated HIF-1 α levels are essential for the reduction in scarring. Collectively, increased HIF-1 α activation upon partial epidermal reprogramming—particularly the atypical pattern observed in the old epidermis near the initial wound site rather than in the newly generated epithelium (Fig. 7b-d)—substantially contributes to the widespread changes that promote wound repair.”

Reversion of reduced scarring in Epi-iOSKM skin by HIF-1 α inhibition (new Fig. 7m,n; new Supplementary Fig. 15e)

These results illustrate how epidermal changes can influence dermal healing. HIF-1 α , a responder to stressful environments, is rapidly activated after wounding and coordinates multiple repair processes^{12,21}. When the control group was transiently treated with by a HIF-1 α inhibitor, re-epithelialization was not altered, but dermal healing process was affected (Fig. 7j-n), suggesting that the primary role of HIF-1 α during the early phase of wound healing is to coordinate dermal niche activation. Thus, the alterations in HIF-1 α activity may broadly affect healing processes. Indeed, spatial and temporal regulation of HIF-1 α is known to exert complex, context-dependent effects on healing, which can be either beneficial or detrimental^{14,21,22}.

The Epi-iOSKM epidermis exhibited the elevation of HIF-1 α activity as an unusual pattern: the signal was increased in the old epithelium that had undergone partial reprogramming, rather than in the newly regenerated epithelium (Fig. 7b-d).

HIF-1 α activation in wounded Epi-iOSKM epidermis (Fig. 7b-d)

This alteration substantially remodeled wound repair mechanisms, including re-epithelialization, angiogenesis, and scarring (Fig. 9b). Given the complexity of the changes, it is challenging to establish direct causality among the different repair processes. Nevertheless, our results suggest the potential that accelerated re-epithelialization, accompanied by reduced neovascularization near the rapidly regenerating epithelium, contributes to decreased scar formation. Importantly, HIF-1 α emerges as a key regulator coordinating these processes. Taken together, partial epidermal reprogramming, which induces healing characteristics, activated HIF-1 α even prior to wounding (Fig. 2l), and wounding itself also activated HIF-1 α . Consequently, the Epi-iOSKM epidermis during healing exhibited an atypical pattern of HIF-1 α activation, which appears to promote coordinated repair of both the epidermis and dermis.

Q27. Title of Fig. 1: The reprogramming does not drive epidermal dedifferentiation – it prevents differentiation (the K14 promoter targets non-differentiated basal cells).

Thank you for the comments. The loss of differentiated characteristics in the epidermis can result from a block in the differentiation of basal cells or from the dedifferentiation of suprabasal cells. Among these two possibilities, our data strongly indicate that suprabasal keratinocytes undergo dedifferentiation upon partial epidermal reprogramming (Please see also responses to Reviewer 1, question #2).

Firstly, our system using the Krt14-CreER; LoxP-STOP-LoxP-rtTA-EGFP mouse line can induce rtTA-EGFP expression and following OSKM-mCherry expression in both basal and suprabasal epidermal keratinocytes. This is shown in UMAP plots from scRNA-seq data of Epi-iOSKM mCh^{pos} cells (Fig. 1d; Supplementary Fig. 3d), and in staining results with quantifications of the proportion of Sox2⁺ cells, conducted according to the suggestion in Reviewer 2 question #6 (new Supplementary Fig. 1j). Both results demonstrate mCh^{pos} cells exist both in basal and suprabasal layer. In addition, fluorescent images in Fig. 1b and new Supplementary Fig. 1d show GFP expression in whole epidermal layer, and most of our fluorescent images show mCherry expression in suprabasal keratinocytes (Fig. 1b,c,h,k, etc.).

OSKM induction in both basal and suprabasal layers of Epi-iOSKM epidermis (new Supplementary Fig. 1j)

GFP and mCherry expression in both basal and suprabasal layers of Epi-iOSKM epidermis (new Supplementary Fig. 1d)

This occurs because our system has two steps for ectopic gene expression: rtTA-EGFP expressions in Krt14⁺ cells following tamoxifen injection, and OSKM-mCherry expressions in their progenies under Dox administration.

Two steps to induce OSKM-mCherry expression in our system (Fig. 1a)

We note that the excision of LoxP sites by CreER activation upon tamoxifen administration is irreversible during physiological differentiation. Krt14 promoter targets non-differentiated basal cells, but these cells differentiate into the suprabasal keratinocytes by maintaining the genetic modifications acquired when they were basal epidermal cells. We injected tamoxifen for six days, took a day off, and administrated Dox for three days. Thus, OSKM induction occurred for a maximum of seven to ten days after CreER activation upon tamoxifen treatment. This is enough time to rtTA-expressing basal cells differentiate into suprabasal cells, by considering that it takes 4-5 days for keratinocytes to differentiate from the basal layer to the suprabasal layer, and 8-10 days for them to undergo terminal differentiation and be shed from the epidermis^{64,65}. In conclusion, our system can induce OSKM expression in all epidermal keratinocytes differentiated from Krt14⁺ epidermal stem cells.

We have shown a loss of differentiated characteristics in the Epi-iOSKM epidermis, including reductions in differentiation marker expression and correlated pseudotime values (Fig. 1f-i; Supplementary Fig. 4c,e,f, new Supplementary Fig. 4d,g,h). We also identified that partial epidermal reprogramming induces plasticity within epidermal cell lineages (Fig. 2; Supplementary Fig. 8). In addition to these, our original and new data from RNA velocity analyses clearly demonstrate that these findings reflect dedifferentiation, not the perturbation of differentiation. Specifically, mCh^{pos} and mCh^{neg} cells of the Epi-iOSKM epidermis exhibited cell fate transitions from Supra to Basal IFE cell, showing an opposite pattern to the control cells, which displayed a differentiation pathway from Basal to Supra cells (Fig. 1l).

Moreover, we provide new consistent data with marker gene expressions. In control cells, the trajectory is positively correlated with the increase of epidermal differentiation marker *Krt10* and *Dsg1a* expression, and the decrease of epidermal stem cell marker *Krt14* and *Itga6* expression. However, Epi-iOSKM cells exhibited the opposite pattern. The partially reprogrammed cells and their neighbors were transitioned from epidermal differentiation marker (*Krt10*, *Dsg1a*)-highly expressing cells to epidermal stem cell marker (*Krt14*, *Itga6*)-highly expressing cells, representing a reversal of the normal differentiation process (new Fig.1m; new Supplementary Fig. 4l-n). These data demonstrate the partial dedifferentiation of epidermal keratinocytes within their lineages in cell-autonomous and non-cell-autonomous manners.

Cell fate transition from Supra to Basal cells under partial reprogramming (Fig. 1l)

Cell fate transition from differentiated to stem-like IFE cells under partial reprogramming (new Fig. 1m; new Supplementary Fig. 4l-n)

References

- 1 Kurian, L. *et al.* Conversion of human fibroblasts to angioblast-like progenitor cells. *Nat Methods* **10**, 77-83 (2013).
- 2 Thier, M. *et al.* Direct conversion of fibroblasts into stably expandable neural stem cells. *Cell Stem Cell* **10**, 473-479 (2012).
- 3 Ocampo, A. *et al.* In Vivo Amelioration of Age-Associated Hallmarks by Partial Reprogramming. *Cell* **167**, 1719-1733 e1712 (2016).
- 4 Browder, K. C. *et al.* In vivo partial reprogramming alters age-associated molecular changes during physiological aging in mice. *Nat Aging* **2**, 243-253 (2022).
- 5 Wang, C. *et al.* In vivo partial reprogramming of myofibers promotes muscle regeneration by remodeling the stem cell niche. *Nat Commun* **12**, 3094 (2021).
- 6 Chen, Y. *et al.* Reversible reprogramming of cardiomyocytes to a fetal state drives heart regeneration in mice. *Science* **373**, 1537-1540 (2021).
- 7 Kim, J. *et al.* Partial in vivo reprogramming enables injury-free intestinal regeneration via autonomous Ptgs1 induction. *Sci Adv* **9**, eadi8454 (2023).
- 8 Yucel, A. D. & Gladyshev, V. N. The long and winding road of reprogramming-induced rejuvenation. *Nat Commun* **15**, 1941 (2024).
- 9 Cohen, E. *et al.* Significance of stress keratin expression in normal and diseased epithelia. *iScience* **27**, 108805 (2024).
- 10 Hasche, D. *et al.* Cytokeratin 17 expression is commonly observed in keratinocytic skin tumours and controls tissue homeostasis impacting human papillomavirus protein expression. *Br J Dermatol* **191**, 949-963 (2024).
- 11 Depianto, D., Kerns, M. L., Dlugosz, A. A. & Coulombe, P. A. Keratin 17 promotes epithelial proliferation and tumor growth by polarizing the immune response in skin. *Nat Genet* **42**, 910-914 (2010).
- 12 Liu, S. *et al.* A tissue injury sensing and repair pathway distinct from host pathogen defense. *Cell* **186**, 2127-2143 e2122 (2023).
- 13 Gibbs, S. *et al.* Epidermal growth factor and keratinocyte growth factor differentially

- regulate epidermal migration, growth, and differentiation. *Wound Repair Regen* **8**, 192-203 (2000).
- 14 Elson, D. A., Ryan, H. E., Snow, J. W., Johnson, R. & Arbeit, J. M. Coordinate up-regulation of hypoxia inducible factor (HIF)-1alpha and HIF-1 target genes during multi-stage epidermal carcinogenesis and wound healing. *Cancer Res* **60**, 6189-6195 (2000).
- 15 Gallant-Behm, C. L. & Mustoe, T. A. Occlusion regulates epidermal cytokine production and inhibits scar formation. *Wound Repair Regen* **18**, 235-244 (2010).
- 16 Cubison, T. C., Pape, S. A. & Parkhouse, N. Evidence for the link between healing time and the development of hypertrophic scars (HTS) in paediatric burns due to scald injury. *Burns* **32**, 992-999 (2006).
- 17 Finnerty, C. C. *et al.* Hypertrophic scarring: the greatest unmet challenge after burn injury. *Lancet* **388**, 1427-1436 (2016).
- 18 Ring, N. A. R. *et al.* The p-rpS6-zone delineates wounding responses and the healing process. *Dev Cell* **58**, 981-992 e986 (2023).
- 19 Doeser, M. C., Scholer, H. R. & Wu, G. Reduction of Fibrosis and Scar Formation by Partial Reprogramming In Vivo. *Stem Cells* **36**, 1216-1225 (2018).
- 20 Reynolds, L. E. *et al.* Accelerated re-epithelialization in beta3-integrin-deficient- mice is associated with enhanced TGF-beta1 signaling. *Nat Med* **11**, 167-174 (2005).
- 21 Li, G. *et al.* A small molecule HIF-1alpha stabilizer that accelerates diabetic wound healing. *Nat Commun* **12**, 3363 (2021).
- 22 Tai, Y., Zheng, L., Liao, J., Wang, Z. & Zhang, L. Roles of the HIF-1alpha pathway in the development and progression of keloids. *Heliyon* **9**, e18651 (2023).
- 23 Joost, S. *et al.* Single-Cell Transcriptomics of Traced Epidermal and Hair Follicle Stem Cells Reveals Rapid Adaptations during Wound Healing. *Cell Rep* **25**, 585-597 e587 (2018).
- 24 Lim, C., Lim, J. & Choi, S. Wound-Induced Hair Follicle Neogenesis as a Promising

- Approach for Hair Regeneration. *Mol Cells* **46**, 573-578 (2023).
- 25 Zhang, Y. V., White, B. S., Shalloway, D. I. & Tumber, T. Stem cell dynamics in mouse hair follicles: a story from cell division counting and single cell lineage tracing. *Cell Cycle* **9**, 1504-1510 (2010).
- 26 Mascré, G. *et al.* Distinct contribution of stem and progenitor cells to epidermal maintenance. *Nature* **489**, 257-262 (2012).
- 27 Villarreal-Ponce, A. *et al.* Keratinocyte-Macrophage Crosstalk by the Nrf2/Ccl2/EGF Signaling Axis Orchestrates Tissue Repair. *Cell Rep* **33**, 108417 (2020).
- 28 Bhattacharya, S. *et al.* DLX3-Dependent STAT3 Signaling in Keratinocytes Regulates Skin Immune Homeostasis. *J Invest Dermatol* **138**, 1052-1061 (2018).
- 29 Sheng, C. *et al.* A stably self-renewing adult blood-derived induced neural stem cell exhibiting patternability and epigenetic rejuvenation. *Nat Commun* **9**, 4047 (2018).
- 30 Onder, T. T. *et al.* Chromatin-modifying enzymes as modulators of reprogramming. *Nature* **483**, 598-602 (2012).
- 31 Wei, Z. *et al.* Klf4 interacts directly with Oct4 and Sox2 to promote reprogramming. *Stem Cells* **27**, 2969-2978 (2009).
- 32 Lu, Y. *et al.* Reprogramming to recover youthful epigenetic information and restore vision. *Nature* **588**, 124-129 (2020).
- 33 Woodley, D. T. *et al.* Intra-dermal injection of lentiviral vectors corrects regenerated human dystrophic epidermolysis bullosa skin tissue in vivo. *Mol Ther* **10**, 318-326 (2004).
- 34 Kuhn, U., Terunuma, A., Pflutzner, W., Foster, R. A. & Vogel, J. C. In vivo assessment of gene delivery to keratinocytes by lentiviral vectors. *J Virol* **76**, 1496-1504 (2002).
- 35 Huang, M. J., Cheng, Y. C., Liu, C. R., Lin, S. & Liu, H. E. A small-molecule c-Myc inhibitor, 10058-F4, induces cell-cycle arrest, apoptosis, and myeloid differentiation of human acute myeloid leukemia. *Exp Hematol* **34**, 1480-1489 (2006).
- 36 Zomer, H. D. & Trentin, A. G. Skin wound healing in humans and mice: Challenges in

- translational research. *J Dermatol Sci* **90**, 3-12 (2018).
- 37 Mascharak, S. *et al.* Preventing Engrailed-1 activation in fibroblasts yields wound regeneration without scarring. *Science* **372** (2021).
- 38 Mack, K. L. *et al.* Allele-specific expression reveals genetic drivers of tissue regeneration in mice. *Cell Stem Cell* **30**, 1368-1381 e1366 (2023).
- 39 Pignet, A. L., Schellnegger, M., Hecker, A., Kamolz, L. P. & Kotzbeck, P. Modeling Wound Chronicity In Vivo: The Translational Challenge to Capture the Complexity of Chronic Wounds. *J Invest Dermatol* **144**, 1454-1470 (2024).
- 40 Xie, Y. *et al.* SHED-derived exosomes promote LPS-induced wound healing with less itching by stimulating macrophage autophagy. *J Nanobiotechnology* **20**, 239 (2022).
- 41 He, Y. *et al.* Tryptanthrin promotes pressure ulcers healing in mice by inhibiting macrophage-mediated inflammation via cGAS/STING pathways. *Int Immunopharmacol* **130**, 111687 (2024).
- 42 Wyles, S. P. *et al.* A chronic wound model to investigate skin cellular senescence. *Aging (Albany NY)* **15**, 2852-2862 (2023).
- 43 Tan, J. L. *et al.* Restoration of the healing microenvironment in diabetic wounds with matrix-binding IL-1 receptor antagonist. *Commun Biol* **4**, 422 (2021).
- 44 Falanga, V. *et al.* Chronic wounds. *Nat Rev Dis Primers* **8**, 50 (2022).
- 45 Ojeh, N. *et al.* The Wound Reporting in Animal and Human Preclinical Studies (WRAHPS) guidelines. *Wounds* **37**, 13-45 (2025).
- 46 Tan, M. L. L., Chin, J. S., Madden, L. & Becker, D. L. Challenges faced in developing an ideal chronic wound model. *Expert Opin Drug Discov* **18**, 99-114 (2023).
- 47 Ma, J. *et al.* Single-cell RNA-Seq analysis of diabetic wound macrophages in STZ-induced mice. *J Cell Commun Signal* **17**, 103-120 (2023).
- 48 Avitabile, S. *et al.* Interleukin-22 Promotes Wound Repair in Diabetes by Improving Keratinocyte Pro-Healing Functions. *J Invest Dermatol* **135**, 2862-2870 (2015).
- 49 Sun, Y., Song, L., Zhang, Y., Wang, H. & Dong, X. Adipose stem cells from type 2

- diabetic mice exhibit therapeutic potential in wound healing. *Stem Cell Res Ther* **11**, 298 (2020).
- 50 Okizaki, S. *et al.* Vascular Endothelial Growth Factor Receptor Type 1 Signaling Prevents Delayed Wound Healing in Diabetes by Attenuating the Production of IL-1beta by Recruited Macrophages. *Am J Pathol* **186**, 1481-1498 (2016).
- 51 Mieczkowski, M. *et al.* Insulin, but Not Metformin, Supports Wound Healing Process in Rats with Streptozotocin-Induced Diabetes. *Diabetes Metab Syndr Obes* **14**, 1505-1517 (2021).
- 52 Raimondi, A. R., Molinolo, A. & Gutkind, J. S. Rapamycin prevents early onset of tumorigenesis in an oral-specific K-ras and p53 two-hit carcinogenesis model. *Cancer Res* **69**, 4159-4166 (2009).
- 53 Youssef, K. K. *et al.* Identification of the cell lineage at the origin of basal cell carcinoma. *Nat Cell Biol* **12**, 299-305 (2010).
- 54 Elbaz, J. *et al.* Highly efficient reprogrammable mouse lines with integrated reporters to track the route to pluripotency. *Proc Natl Acad Sci U S A* **119**, e2207824119 (2022).
- 55 Matsumura, S., Terao, M., Murota, H. & Katayama, I. Th2 cytokines enhance TrkA expression, upregulate proliferation, and downregulate differentiation of keratinocytes. *J Dermatol Sci* **78**, 215-223 (2015).
- 56 Yates, C. C. *et al.* Delayed and deficient dermal maturation in mice lacking the CXCR3 ELR-negative CXC chemokine receptor. *Am J Pathol* **171**, 484-495 (2007).
- 57 Xu, F. *et al.* CXCL10 secreted by SPRY1-deficient epidermal keratinocytes fuels joint inflammation in psoriatic arthritis via CD14 signaling. *J Clin Invest* **135** (2025).
- 58 Shibuya, R. *et al.* CCL2–CCR2 Signaling in the Skin Drives Surfactant-Induced Irritant Contact Dermatitis through IL-1beta–Mediated Neutrophil Accumulation. *J Invest Dermatol* **142**, 571-582 e579 (2022).
- 59 Arafa, E. *et al.* TMIGD1 is a novel adhesion molecule that protects epithelial cells from oxidative cell injury. *Am J Pathol* **185**, 2757-2767 (2015).

- 60 Wu, Y. *et al.* TMIGD1 Inhibited Abdominal Adhesion Formation by Alleviating Oxidative Stress in the Mitochondria of Peritoneal Mesothelial Cells. *Oxid Med Cell Longev* **2021**, 9993704 (2021).
- 61 Das, A. T., Tenenbaum, L. & Berkhout, B. Tet-On Systems For Doxycycline-inducible Gene Expression. *Curr Gene Ther* **16**, 156-167 (2016).
- 62 Ge, Y. *et al.* Stem Cell Lineage Infidelity Drives Wound Repair and Cancer. *Cell* **169**, 636-650 e614 (2017).
- 63 Liu, Z. *et al.* Spatiotemporal single-cell roadmap of human skin wound healing. *Cell Stem Cell* **32**, 479-498 e478 (2025).
- 64 Potten, C. S., Saffhill, R. & Maibach, H. I. Measurement of the transit time for cells through the epidermis and stratum corneum of the mouse and guinea-pig. *Cell Tissue Kinet* **20**, 461-472 (1987).
- 65 Zhu, X. *et al.* HDAC1/2 Control Proliferation and Survival in Adult Epidermis and Pre-Basal Cell Carcinoma through p16 and p53. *J Invest Dermatol* **142**, 77-87 e10 (2022).
- 66 Lin, Y., Zhang, W., Li, B. & Wang, G. Keratin 17 in psoriasis: Current understanding and future perspectives. *Semin Cell Dev Biol* **128**, 112-119 (2022).
- 67 Chen, H. *et al.* AKT and its related molecular feature in aged mice skin. *PLoS One* **12**, e0178969 (2017).
- 68 Perez White, B. E. *et al.* Receptor Tyrosine Kinase EPHA2 Drives Epidermal Differentiation through Regulation of EGFR Signaling. *J Invest Dermatol* **144**, 1798-1807 e1791 (2024).
- 69 Dai, X. *et al.* Nuclear IL-33 Plays an Important Role in EGFR-Mediated Keratinocyte Migration by Regulating the Activation of Signal Transducer and Activator of Transcription 3 and NF-kappaB. *JID Innov* **3**, 100205 (2023).
- 70 Boix, J. *et al.* Constitutive HIF-1alpha Expression in the Epidermis Fuels Proliferation and Is Essential for Effective Barrier Formation. *J Invest Dermatol* **145**, 1683-1692 e1688 (2025).

- 71 Kim, J. H. *et al.* HIF-1alpha-mediated BMP6 down-regulation leads to hyperproliferation and abnormal differentiation of keratinocytes in vitro. *Exp Dermatol* **27**, 1287-1293 (2018).
- 72 Jacob, J. T. *et al.* Keratin 17 regulates nuclear morphology and chromatin organization. *J Cell Sci* **133** (2020).
- 73 Guo, S. *et al.* Hypoxia-induced RHCG as a key regulator in psoriasis and its modulation by secukinumab. *APL Bioeng* **9**, 026115 (2025).
- 74 Subudhi, I. *et al.* Metabolic coordination between skin epithelium and type 17 immunity sustains chronic skin inflammation. *Immunity* **57**, 1665-1680 e1667 (2024).
- 75 Shi, X. *et al.* IL-17A upregulates keratin 17 expression in keratinocytes through STAT1- and STAT3-dependent mechanisms. *J Invest Dermatol* **131**, 2401-2408 (2011).
- 76 Mei, L. & Nave, K. A. Neuregulin-ERBB signaling in the nervous system and neuropsychiatric diseases. *Neuron* **83**, 27-49 (2014).
- 77 Roskoski, R., Jr. The ErbB/HER family of protein-tyrosine kinases and cancer. *Pharmacol Res* **79**, 34-74 (2014).
- 78 van Lengerich, B., Agnew, C., Puchner, E. M., Huang, B. & Jura, N. EGF and NRG induce phosphorylation of HER3/ERBB3 by EGFR using distinct oligomeric mechanisms. *Proc Natl Acad Sci U S A* **114**, E2836-E2845 (2017).
- 79 Uliano, J., Corvaja, C., Curigliano, G. & Tarantino, P. Targeting HER3 for cancer treatment: a new horizon for an old target. *ESMO Open* **8**, 100790 (2023).
- 80 Han, W. & He, Y. Y. Requirement for metalloproteinase-dependent ERK and AKT activation in UVB-induced G1-S cell cycle progression of human keratinocytes. *Photochem Photobiol* **85**, 997-1003 (2009).
- 81 Young, C. D. *et al.* Activating PIK3CA Mutations Induce an Epidermal Growth Factor Receptor (EGFR)/Extracellular Signal-regulated Kinase (ERK) Paracrine Signaling Axis in Basal-like Breast Cancer. *Mol Cell Proteomics* **14**, 1959-1976 (2015).
- 82 Baumer, A. T. *et al.* Phosphatidylinositol 3-kinase-dependent membrane recruitment

- of Rac-1 and p47phox is critical for alpha-platelet-derived growth factor receptor-induced production of reactive oxygen species. *J Biol Chem* **283**, 7864-7876 (2008).
- 83 Peus, D. *et al.* H₂O₂ is an important mediator of UVB-induced EGF-receptor phosphorylation in cultured keratinocytes. *J Invest Dermatol* **110**, 966-971 (1998).
- 84 Chiu, L. Y. *et al.* PARP-1 involves in UVB-induced inflammatory response in keratinocytes and skin injury via regulation of ROS-dependent EGFR transactivation and p38 signaling. *FASEB J* **35**, e21393 (2021).
- 85 Werner, S. & Smola, H. Paracrine regulation of keratinocyte proliferation and differentiation. *Trends Cell Biol* **11**, 143-146 (2001).
- 86 Groves, R. W. *et al.* Inflammatory and hyperproliferative skin disease in mice that express elevated levels of the IL-1 receptor (type I) on epidermal keratinocytes. Evidence that IL-1-inducible secondary cytokines produced by keratinocytes in vivo can cause skin disease. *J Clin Invest* **98**, 336-344 (1996).
- 87 Cai, Y. *et al.* A Critical Role of the IL-1beta-IL-1R Signaling Pathway in Skin Inflammation and Psoriasis Pathogenesis. *J Invest Dermatol* **139**, 146-156 (2019).
- 88 Maas-Szabowski, N., Shimotoyodome, A. & Fusenig, N. E. Keratinocyte growth regulation in fibroblast cocultures via a double paracrine mechanism. *J Cell Sci* **112 (Pt 12)**, 1843-1853 (1999).
- 89 Wilgus, T. A. *et al.* Novel function for vascular endothelial growth factor receptor-1 on epidermal keratinocytes. *Am J Pathol* **167**, 1257-1266 (2005).
- 90 Brown, L. F. *et al.* Expression of vascular permeability factor (vascular endothelial growth factor) by epidermal keratinocytes during wound healing. *J Exp Med* **176**, 1375-1379 (1992).
- 91 Man, X. Y., Yang, X. H., Cai, S. Q., Bu, Z. Y. & Zheng, M. Overexpression of vascular endothelial growth factor (VEGF) receptors on keratinocytes in psoriasis: regulated by calcium independent of VEGF. *J Cell Mol Med* **12**, 649-660 (2008).
- 92 Sharma, M., Chuang, W. W. & Sun, Z. Phosphatidylinositol 3-kinase/Akt stimulates

- androgen pathway through GSK3beta inhibition and nuclear beta-catenin accumulation. *J Biol Chem* **277**, 30935-30941 (2002).
- 93 Mayer, I. A. & Arteaga, C. L. The PI3K/AKT Pathway as a Target for Cancer Treatment. *Annu Rev Med* **67**, 11-28 (2016).
- 94 Flanagan, D. J., Woodcock, S. A., Phillips, C., Eagle, C. & Sansom, O. J. Targeting ligand-dependent wnt pathway dysregulation in gastrointestinal cancers through porcupine inhibition. *Pharmacol Ther* **238**, 108179 (2022).
- 95 Koo, B. K., van Es, J. H., van den Born, M. & Clevers, H. Porcupine inhibitor suppresses paracrine Wnt-driven growth of Rnf43;Znrf3-mutant neoplasia. *Proc Natl Acad Sci U S A* **112**, 7548-7550 (2015).
- 96 Han, T. *et al.* R-Spondin chromosome rearrangements drive Wnt-dependent tumour initiation and maintenance in the intestine. *Nat Commun* **8**, 15945 (2017).
- 97 Havran, W. L. & Allison, J. P. Origin of Thy-1+ dendritic epidermal cells of adult mice from fetal thymic precursors. *Nature* **344**, 68-70 (1990).
- 98 Konieczny, P. *et al.* Interleukin-17 governs hypoxic adaptation of injured epithelium. *Science* **377**, eabg9302 (2022).
- 99 Anderson, L. S. *et al.* CCR6(+) gammadelta T Cells Home to Skin Wounds and Restore Normal Wound Healing in CCR6-Deficient Mice. *J Invest Dermatol* **139**, 2061-2064 e2062 (2019).
- 100 Lawler, W. *et al.* Impact of obesity on the CCR6-CCL20 axis in epidermal gammadelta T cells and IL-17A production in murine wound healing and psoriasis. *J Immunol* **214**, 153-166 (2025).
- 101 Kennedy-Crispin, M. *et al.* Human keratinocytes' response to injury upregulates CCL20 and other genes linking innate and adaptive immunity. *J Invest Dermatol* **132**, 105-113 (2012).
- 102 Martin, B., Hirota, K., Cua, D. J., Stockinger, B. & Veldhoen, M. Interleukin-17-producing gammadelta T cells selectively expand in response to pathogen products

- and environmental signals. *Immunity* **31**, 321-330 (2009).
- 103 Hishida, T. *et al.* In vivo partial cellular reprogramming enhances liver plasticity and regeneration. *Cell Rep* **39**, 110730 (2022).
- 104 Abad, M. *et al.* Reprogramming in vivo produces teratomas and iPS cells with totipotency features. *Nature* **502**, 340-345 (2013).
- 105 Parras, A. *et al.* In vivo reprogramming leads to premature death linked to hepatic and intestinal failure. *Nat Aging* **3**, 1509-1520 (2023).
- 106 Naik, S. *et al.* Inflammatory memory sensitizes skin epithelial stem cells to tissue damage. *Nature* **550**, 475-480 (2017).
- 107 Yamamoto, N. & Kiyosawa, T. Histological effects of occlusive dressing on healing of incisional skin wounds. *Int Wound J* **11**, 616-621 (2014).
- 108 Eming, S. A., Wynn, T. A. & Martin, P. Inflammation and metabolism in tissue repair and regeneration. *Science* **356**, 1026-1030 (2017).
- 109 Lucas, T. *et al.* Differential roles of macrophages in diverse phases of skin repair. *J Immunol* **184**, 3964-3977 (2010).
- 110 Mirza, R., DiPietro, L. A. & Koh, T. J. Selective and specific macrophage ablation is detrimental to wound healing in mice. *Am J Pathol* **175**, 2454-2462 (2009).
- 111 Hur, Y. H. Epidermal stem cells: Interplay with the skin microenvironment during wound healing. *Mol Cells* **47**, 100138 (2024).
- 112 Shi, Z. *et al.* Microglia drive transient insult-induced brain injury by chemotactic recruitment of CD8(+) T lymphocytes. *Neuron* **111**, 696-710 e699 (2023).
- 113 Christensen, J. E., de Lemos, C., Moos, T., Christensen, J. P. & Thomsen, A. R. CXCL10 is the key ligand for CXCR3 on CD8+ effector T cells involved in immune surveillance of the lymphocytic choriomeningitis virus-infected central nervous system. *J Immunol* **176**, 4235-4243 (2006).
- 114 Matsumura, S. *et al.* Radiation-induced CXCL16 release by breast cancer cells attracts effector T cells. *J Immunol* **181**, 3099-3107 (2008).

115 Hinz, B. *et al.* The myofibroblast: one function, multiple origins. *Am J Pathol* **170**, 1807-1816 (2007).

REVIEWERS' COMMENTS

Reviewer #2 (Remarks to the Author):

The authors performed a very extensive revision, and the revised manuscript is clearly improved.

Q1, Q2 and Q4 of reviewer 1 have been appropriately addressed, mainly by clarifying that epidermal cells undergo only a partial reprogramming, by further analysis of sc-RNA-seq data, and by further analysis of the hair follicles.

Q3 of reviewer 1 was partially addressed. The HIF1a inhibitor studies strongly suggest that HIF1a is required for wound angiogenesis (as expected from the literature) and also reduces dermal thickness. However, this was seen in control and in iOSKM mice. Therefore, the authors cannot conclude that HIF1a inhibition SPECIFICALLY reverses the iOSKM effect. The data on collagen III positive area suggest a specific effect of HIF1a inhibition on the iOSKM effect, but the variability of the data is high. I believe that this point requires further clarification in the manuscript. There are probably additional mechanisms that contribute to the effect of iOSKM on the dermis/granulation tissue.

It is difficult to explain how enhanced reepithelialization inhibits angiogenesis. Usually, it is the opposite, because keratinocytes are an important source of VEGF. This point should also be discussed.

Taken together, the (non-cell-autonomous) mechanisms underlying the alterations in dermal repair remain speculative, and therefore, this part of the study remains descriptive. This limitation should be mentioned in the discussion.

The questions of reviewer 2 have been appropriately addressed.

We sincerely thank Reviewer 2 for the careful and thorough evaluation of the revised manuscript, including the reassessment of the comments originally raised by Reviewer 1. We are pleased that the reviewer finds the manuscript to be substantially improved and that most concerns have been satisfactorily addressed. Below, we provide detailed responses to the remaining points, particularly those related to the interpretation of the HIF-1 α experiments and dermal repair mechanisms.

Specificity of HIF-1 α inhibition in Epi-iOSKM skin

We agree with the reviewer's assessment and have revised the manuscript to clarify this point. Our data indicate that HIF-1 α inhibition affects dermal repair in both control and Epi-iOSKM skin, demonstrating that HIF-1 α functions as a general regulator of wound healing rather than a factor acting exclusively under partial reprogramming conditions.

Accordingly, we have revised our conclusions and now describe HIF-1 α as an essential mediator of enhanced dermal repair in Epi-iOSKM skin, but not a reprogramming-specific factor. While HIF-1 α inhibition reversed the enhanced dermal healing phenotypes observed in Epi-iOSKM skin, similar effects were also observed in control skin (Fig. 7k–n). This point is now clearly stated in the revised Discussion.

Relationship between enhanced re-epithelialization and altered angiogenesis

We appreciate this insightful comment and have expanded the Discussion to directly address this apparent paradox. In the revised manuscript, we propose that the altered angiogenesis pattern in Epi-iOSKM skin may be explained not by reduced pro-angiogenic signaling itself, but by the spatial distribution of HIF-1 α activation.

Specifically, HIF-1 α activation was predominantly detected in the pre-existing epidermis undergoing partial reprogramming, rather than in the newly regenerated epidermis at the leading edge (Fig. 7b–d). Given that HIF-1 α regulates angiogenic and chemotactic signaling pathways^{1,2}, stronger signaling near the initial wound site could bias neovascularization toward this region and limit vessel extension toward the migrating epidermal front.

We now clearly state that this explanation remains hypothetical and that a direct causal link has not yet been demonstrated.

Speculative and descriptive nature of non-cell-autonomous dermal mechanisms

We agree with this assessment and have revised the Discussion accordingly. We now explicitly state that, although our study identifies HIF-1 α as one of the key mediators linking epidermal partial reprogramming to altered dermal repair, the underlying non-cell-autonomous mechanisms remain incompletely defined.

We emphasize that this aspect of the study is descriptive and hypothesis-generating, and we outline future studies aimed at dissecting how spatially and temporally restricted activation of epidermal stress-response pathways influences dermal repair and scar formation.

In response to the reviewer's comments, we added the following clarification to the Discussion section:

Mosaic partial reprogramming in IFE cells promoted not only epidermal repair but also dermal repair. In particular, accelerated re-epithelialization was accompanied by altered angiogenesis patterns and reduced scar formation. Mechanistically, we identified epidermal HIF-1 α activation as a key factor contributing to enhanced dermal healing in Epi-iOSKM skin. However, both enhanced re-epithelialization and HIF-1 α activation are generally associated with increased angiogenesis, contrary to the angiogenic phenotype observed in Epi-iOSKM skin. One possible explanation for this unexpected angiogenesis pattern is the spatially biased hyperactivation of HIF-1 α . Specifically, HIF-1 α signaling was increased in the old epithelium that had undergone partial reprogramming, rather than in the newly regenerated epithelium (Fig. 7b-d). Because HIF-1 α regulates angiogenic and chemotactic signaling pathways^{1,2}, relatively stronger HIF-1 α induction near the initial wound site compared to the leading edge may contribute to a spatial bias in neovascularization, with enhanced vascularization in the initial wound-adjacent dermis and reduced vessel extension toward the leading edge. Although this model is plausible, a direct causal relationship has not been demonstrated. To assess the contribution of HIF-1 α to the repair phenotypes of Epi-iOSKM skin, we inhibited HIF-1 α activity, which reversed the enhanced dermal healing phenotype. However, similar effects were also observed in control skin (Fig. 7k-n), indicating that HIF-1 α does not function exclusively in the context of partial reprogramming. Indeed, HIF-1 α is a stress-responsive transcription factor that is rapidly activated after wounding and coordinates multiple repair processes^{3,4}. Taken together, our data identify HIF-1 α as one of the key factors linking the effects of partial epidermal reprogramming to altered dermal healing. However, the precise mechanisms by which the transient activation of IFE cells leads to reduced scar formation remain to be elucidated.

References

- 1 Pugh, C. W. & Ratcliffe, P. J. Regulation of angiogenesis by hypoxia: role of the HIF system. *Nat Med* **9**, 677-684 (2003).
- 2 Forsythe, J. A. *et al.* Activation of vascular endothelial growth factor gene transcription by hypoxia-inducible factor 1. *Mol Cell Biol* **16**, 4604-4613 (1996).
- 3 Liu, S. *et al.* A tissue injury sensing and repair pathway distinct from host pathogen defense. *Cell* **186**, 2127-2143 e2122 (2023).

- 4 Li, G. *et al.* A small molecule HIF-1alpha stabilizer that accelerates diabetic wound healing. *Nat Commun* **12**, 3363 (2021).